# Spectral cues are necessary to encode azimuthal auditory space in the mouse superior colliculus

Shinya Ito [1✉], Yufei Si [2], David A. Feldheim[2,3] & Alan M. Litke [1,3✉]

Sound localization plays a critical role in animal survival. Three cues can be used to compute sound direction: interaural timing differences (ITDs), interaural level differences (ILDs) and the direction-dependent spectral filtering by the head and pinnae (spectral cues). Little is known about how spectral cues contribute to the neural encoding of auditory space. Here we report on auditory space encoding in the mouse superior colliculus (SC). We show that the mouse SC contains neurons with spatially-restricted receptive fields (RFs) that form an azimuthal topographic map. We found that frontal RFs require spectral cues and lateral RFs require ILDs. The neurons with frontal RFs have frequency tunings that match the spectral structure of the specific head and pinna filter for sound coming from the front. These results demonstrate that patterned spectral cues in combination with ILDs give rise to the topographic map of azimuthal auditory space.

[1] Santa Cruz Institute for Particle Physics, University of California, Santa Cruz, 1156 High Street, Santa Cruz, CA 95064, USA. [2] The Department of Molecular, Cell and Developmental Biology, University of California, Santa Cruz, 1156 High Street, Santa Cruz, CA 95064, USA. [3]These authors jointly supervised this work: David A. Feldheim, Alan M. Litke. ✉email: sito@ucsc.edu; alan.litke@cern.ch

Sound localization is the result of the sophisticated analysis of vibrations detected by an animal's ears. Unlike vision or touch, where the receptor position along the sensory detector encodes spatial information, the incident direction of the sound source must be computed. This calculation is based on three cues: interaural timing differences (ITDs), interaural level differences (ILDs), and the spectral modification of the sound as it enters the ear (spectral cues). Since Lord Rayleigh's seminal finding that ITDs and ILDs are used for low- and high-frequency sound localization, respectively (the duplex theory)[1], ITDs and ILDs have been considered to be the two primary cues used for sound localization in the horizontal plane (except for small mammals that do not use ITDs[2–4]). On the other hand, spectral cues were hypothesized to be used for resolving directions that the ITDs and ILDs cannot distinguish, namely, for resolving the front–back ambiguity and for determining the sound source elevation[3]. However, it is clear that spectral cues can also be used for horizontal-plane sound localization because ferrets and blind humans can be trained to perform a horizontal sound localization task with only one ear (i.e., without ITDs or ILDs)[5,6]. Therefore, we hypothesized that spectral cues also contribute to azimuthal sound localization.

To determine the mechanisms used to compute sound source direction, we conducted an electrophysiological study in the mouse superior colliculus (SC, also known as the optic tectum (OT) in non-mammalian species). We have chosen to study the SC because it contains spatially tuned auditory neurons and, in some species, a topographic map of auditory space has been observed. For example, the properties of the SC/OT auditory map have been reported in barn owls[7,8], cats[9], ferrets[10], and Guinea pigs[11,12], and even with monaural hearing in which the map is observed only when the stimulus sound is near response threshold[13,14]. The SC also contains maps of visual and somatosensory space and is a well-studied area for topographic map formation[15]. Interestingly, a topographic map of auditory space has not been demonstrated in the mouse SC. In the present study, we used the head-related transfer functions (HRTFs) to present virtual auditory space (VAS) stimuli[2,11,16,17] to an alert, head-fixed mouse while recording from a large population of neurons in the SC. We determined the spatial receptive fields (RFs) and the spectro-temporal RFs (STRFs) of auditory neurons to determine the topography of the auditory spatial map, measure the relative contribution of ITDs, ILDs, and spectral cues to the encoding of a spatial map, and determine the ST patterns used

to encode the spatial RF. We found that the mouse SC contains a topographic map of auditory space along the azimuthal axis; spectral cues and ILDs are used to compute localized auditory RFs, but ITDs are not; the relative importance of spectral cues and ILDs depends on the azimuthal position of the RF; and the spectral tuning properties of neurons with frontal RFs match well with the spectral structure of the frontal HRTFs. These results demonstrate an unexpectedly important role for spectral cues in the formation of the azimuthal sound map, lending new insights into how mice perform sound localization.

## Results

**Mouse SC neurons form a topographic map of azimuthal space.** To determine the auditory response properties and organizational features of mouse SC neurons, we used VAS stimuli and large-scale silicon probe recordings (Fig. 1a). We first measured a set of HRTFs, the modulation of sound by the physical structure of the head and pinnae, that contains each of the three sound localization cues: ITDs, ILDs, and spectral cues (Supplementary Fig. 1). We used the HRTFs to filter a 100-ms white noise burst stimulus and played it via calibrated earphones so that the mouse would hear the sound as if it came from the direction represented by the corresponding HRTF (see Methods). The stimuli were presented at grid locations of 17 azimuths (−144° to 144°) and 5 elevations (0° to 80°), totaling 85 points in a two-dimensional directional field.

To measure the response properties of the SC neurons, we performed electrophysiology using 256-channel multi-shank silicon probes (Fig. 1b), while the VAS stimuli were presented to mice that were awake and allowed to locomote on a cylindrical treadmill (Fig. 1a). In 20 recordings from 10 mice, we recorded the spiking activity of 3556 neurons (excluding axonal signals; Supplementary Fig. 2) from the deep SC (300–1600 μm from the surface of the SC; Fig. 1b, c). The neurons had a variety of temporal response patterns (Supplementary Fig. 3a–d), and the peak response time for the population showed a bimodal distribution (Supplementary Fig. 3e). Because most (77.5 ± 0.6%, the error was derived assuming a binomial distribution, see Methods) neurons had a peak response earlier than 20 ms after the stimulus onset, we used the spikes with latencies <20 ms for our analysis (see Supplementary Fig. 3 for further discussion). We observed significant ($p < 0.001$, see "Significance test for the auditory responses" in Methods) auditory responses from 56.7 ± 0.8% ($n = 2016$) of all the neurons identified. To determine the

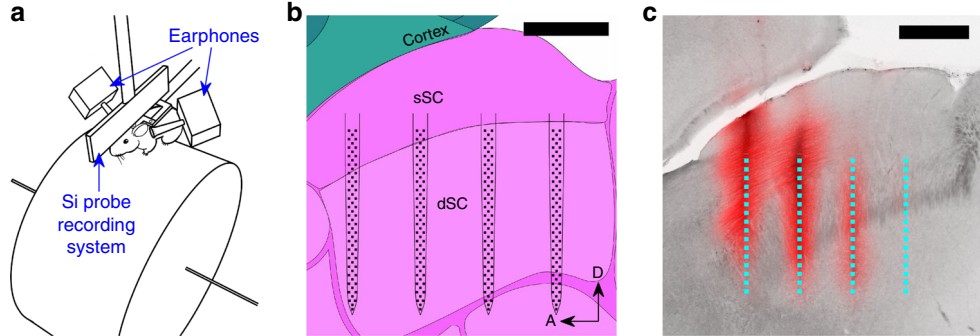

**Fig. 1 Experimental setup. a** An illustration of the virtual auditory space (VAS) stimulus and electrophysiology recording setup. During the recording, the head-fixed mouse can run freely on a cylindrical treadmill, while motion is recorded by a rotary encoder. Auditory stimuli are delivered through a pair of earphones near the mouse's ears. The entire setup is enclosed in an anechoic chamber. **b** A schematic of a 256-electrode silicon probe in the superior colliculus (SC; sagittal plane). The background image is from Allen Mouse Brain Reference Atlas[54]. sSC: superficial SC; dSC: deep SC; A: anterior; D: dorsal. The scale bar is 500 μm. **c** A photograph of a sagittal section of the SC and the fluorescent tracing of DiI (red channel) that was coated on the back of the probe shanks. The dotted cyan lines indicate the active areas of the probe during the recording, reconstructed from the DiI traces (the location of the rightmost shank was inferred by a DiI trace in the adjacent slice). The four shanks with a 400-μm pitch nicely match the anteroposterior extent of the SC. The scale bar is 500 μm.

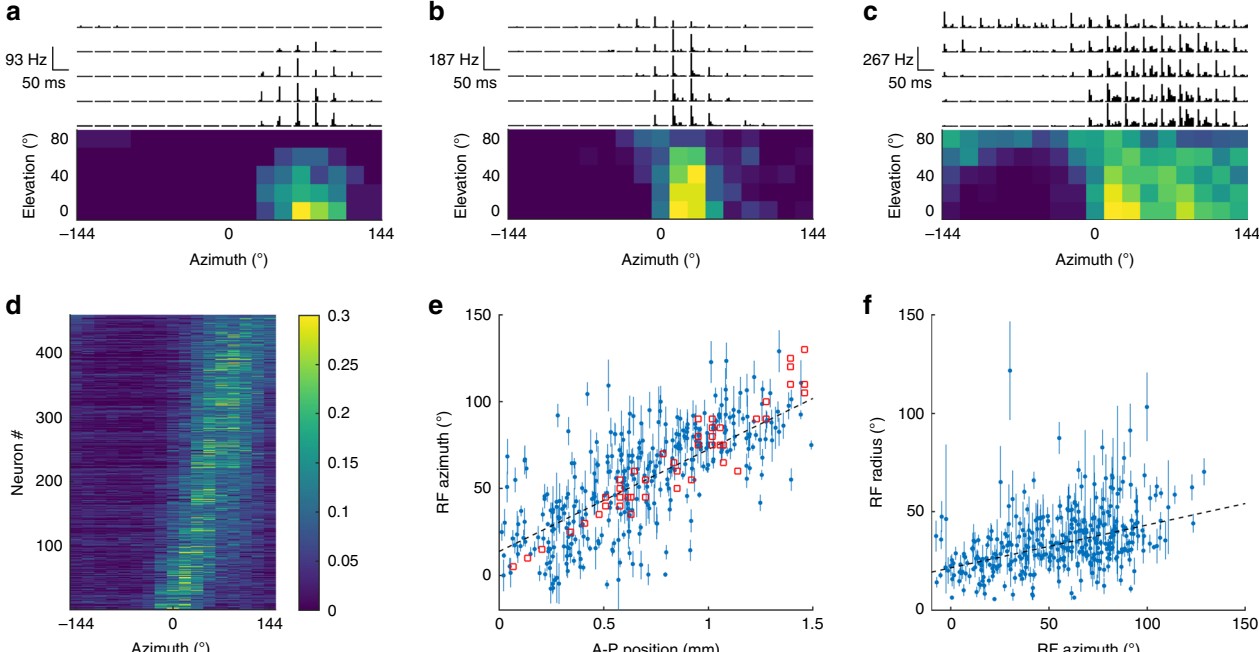

**Fig. 2 Auditory receptive fields (RFs) of SC neurons and their topographic organization. a–c** Examples of the spiking activity and localized RFs of three representative neurons. The top panels are the post-stimulus time histograms (PSTHs) in response to 85 virtual sound source locations. The histograms represent the average firing rate of 30 trials in 5-ms bins. The vertical range is from 0 to the maximum firing rate for the plotted neuron. The bottom panels are summary heatmaps of the activity within the 5–20 ms time interval of the PSTHs. The brighter color indicates a higher firing rate. The RF is visible as a bright area of the figure. **d** Responses of all the neurons that have a localized RF to a stimulus from 17 horizontal directions. For each neuron, the scale of the responses is normalized by dividing by the sum of responses across all directions. Neurons are sorted by their tuned azimuth. **e** A scatter plot of anteroposterior (A–P) SC positions vs. RF azimuths showing topographic organization along the A–P axis of the SC. Each blue dot represents the auditory RF azimuth (center of the Kent distribution) and the A–P position of an individual neuron (the error bars represent the statistical errors derived from the Kent distribution fits; they do not include the systematic errors; see Methods); red squares are visual RFs measured by multi-unit activity in the superficial SC. The slope of the auditory RFs is $58 \pm 4° \, mm^{-1}$ (the black dashed line is the $\chi^2$ fit to the data, including systematic errors; see Methods for details) and that of the visual RFs is $73 \pm 5° \, mm^{-1}$. The offset of the auditory RFs was $14 \pm 3°$. There is a strong correlation between the auditory RF position and the A–P position ($r = 0.70$; Pearson's correlation coefficient). **f** A scatter plot of RF azimuth vs. RF radius of individual neurons. A positive correlation (slope: $0.21 \pm 0.02$; $r = 0.41$) of the RF azimuth and the estimated RF radius (specified by the concentration parameter $\kappa$ of the Kent distribution; see Methods) was observed.

RF properties of these neurons, we parameterized the RFs by fitting the Kent distribution[18] to the directional auditory responses of these neurons. The RFs were considered significant if the Kent distribution explained the data better than a flat distribution (i.e., the Bayesian information criterion of the Kent distribution is smaller than that of a flat distribution; see Methods for details). Of the auditory-responsive neurons, $22.8 \pm 0.9\%$ ($n = 459$) had a significant RF tuned to a specific sound source direction (e.g., Fig. 2a–c and Supplementary Fig. 4).

To determine if the mouse SC contains a topographic map of auditory space, we tested if the location of the RF azimuth and the anteroposterior (A–P) or mediolateral (M–L) position of the neurons in the SC are correlated. We found that the SC neurons have bell-shaped tuning curves with a continuous distribution of preferred azimuths in the horizontal plane (Fig. 2d), with the RF azimuth linearly related to the neurons' A–P location (Fig. 2e). Using line fitting to measure the slopes (considering estimated systematic errors, see Methods), we calculated the slope of the auditory map to be $58 \pm 4° \, mm^{-1}$. This slope is ~20% smaller than the slope measured for the visual map ($p = 0.02$; two-sided analysis of variance; no adjustments), as calculated by multi-unit activity in the superficial SC ($73 \pm 5° \, mm^{-1}$). We also found that the size of a neuron's RF increases as a function of the azimuth (Fig. 2f). The slope calculated between the RF elevations and M–L positions ($13 \pm 5° \, mm^{-1}$) is not as steep as the slope along the azimuth (the slope was not significant in the blinded dataset; see

Supplementary Fig. 5). Comparing the response properties of each neuron when the mouse was stationary vs. running revealed that locomotion increased the spontaneous firing rate and reduced the stimulus-evoked firing rate, but had no influence on the properties of the topographic map (Supplementary Fig. 6). These results demonstrate, for the first time, that the mouse SC contains a topographic map of azimuthal auditory space along the A–P axis of the SC.

**Flattening spectral cues has a large effect on the RFs.** To determine the contribution of each sound localization cue to the formation of spatially restricted RFs, we froze each sound localization cue in the stimuli while leaving the other cues to change naturally, and observed the corresponding change to the RF structure of each neuron. The modified stimulus set included the original stimulus (played twice as a control for assessing the reproducibility of the RFs) as well as the frozen stimuli that had either zero ITDs, zero ILDs, or a flat frequency spectrum, independent of sound direction (see Methods for details). In this experiment, we recorded from 991 neurons from 6 mice, and found that $59.7 \pm 1.6\%$ ($n = 592$) had a significant auditory response and $16.4 \pm 1.5\%$ ($n = 97$) of these had a localized RF. We found that ITD freezing had no effect on the RF structures, consistent with the idea that the distance between the ears of the mouse (~1 cm) is too small to be influenced by the ITD (~29 μs at most). However, the SC neurons changed their RFs by spectrum

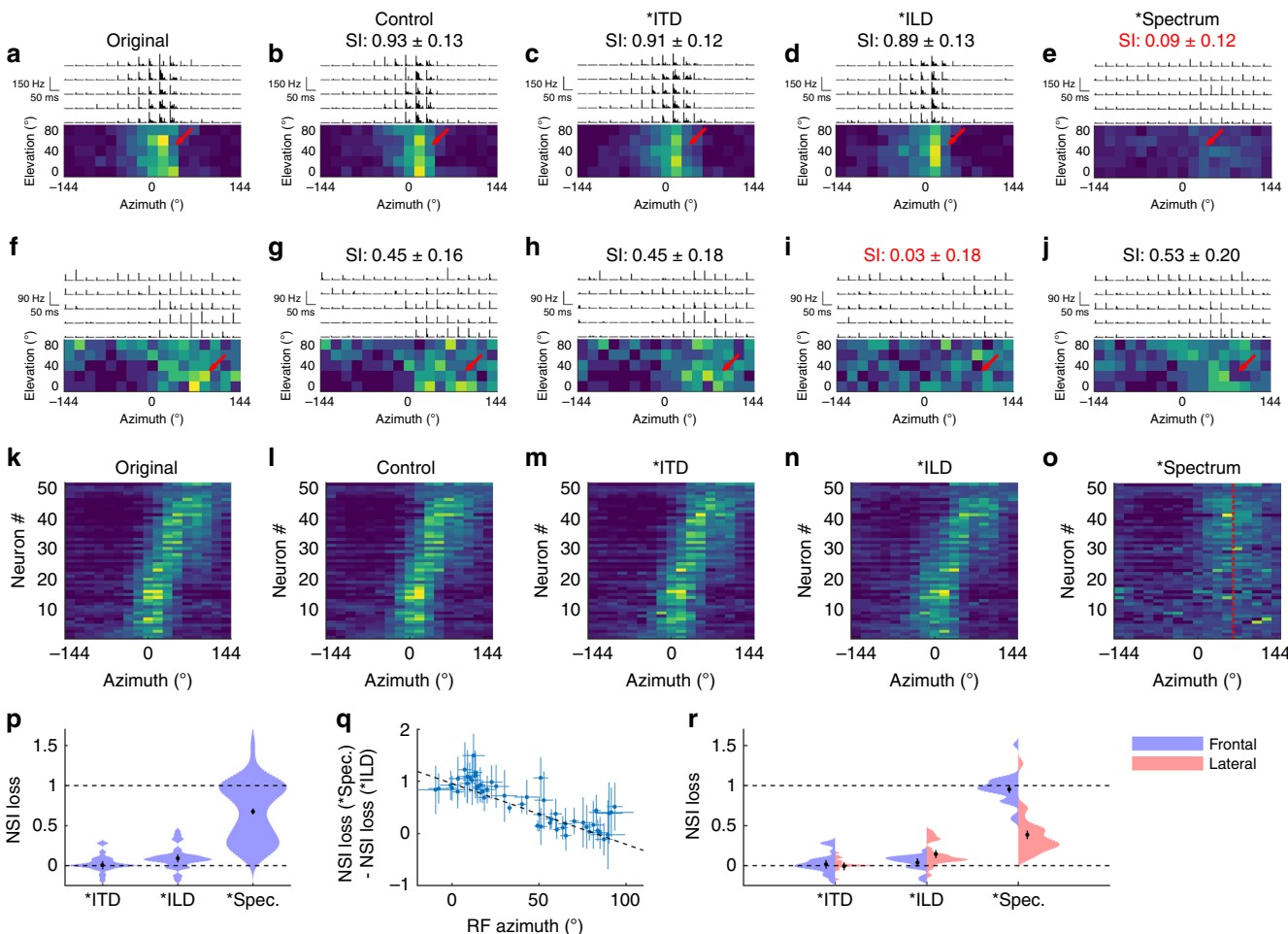

**Fig. 3 Structural changes of auditory RFs due to interaural timing difference (ITD), interaural level difference (ILD), and spectral-cue freezing. a–e** RFs of a single neuron in response to the **a** original, **b** control, **c** frozen ITD, **d** frozen average ILD, and **e** frozen spectrum stimuli. An asterisk (*) indicates that the corresponding cue is frozen. **b–e** indicate the similarity index (SI; see text). This neuron's RF structure did not change during ITD and ILD freezing, but it disappears with spectrum freezing (red arrow). **f–j** Example responses from another neuron whose RF disappears with ILD freezing (red arrow). **k–o** Summary of the horizontal RFs of all 51 neurons that had a reproducible RF. For each neuron, the scale of the responses is normalized by dividing by the sum of responses across all directions. The range of the color scale is from 0 to 0.3. The neurons are sorted by their tuned azimuth. Little changes in the RFs are observed with ITD and ILD freezing (**m**, **n**), but a dramatic change of the RF structure was observed with spectrum freezing (**o**). With spectrum freezing, all the RF centers become close to 65° azimuth where the ILD is maximized (red dotted line). **p** Summary of normalized similarity index (NSI) losses in the cue freezing experiments. A large NSI loss is observed with spectrum freezing, consistent with **m–o**. Source data are provided as a Source Data file. **q** A scatter plot of the RF azimuths and the differences between spectral-cue and ILD NSI losses. As the RF azimuth increases, neurons depend less on spectral cues and more on ILDs ($r = -0.83$). The errors in X-axis are derived from the Kent distribution fits; the errors in Y-axis are based on the Poisson statistics and arithmetic propagation (see Methods). **r** Comparison of the average NSI loss for frontal and lateral RFs. Neurons with frontal rather than lateral RFs depend almost entirely on spectral cues. Source data are provided as a Source Data file.

freezing or ILD freezing, with some affected only by spectrum freezing (Fig. 3a–e), while others were affected only by ILD freezing (Fig. 3f–j).

To quantify the change of the RFs induced by freezing each cue, we used a cosine similarity index (SI). Namely, the response firing rates at 85 spatial locations are considered as a vector, and a normalized inner product was calculated between the responses to the original and the frozen cue stimuli (see Methods for details). If a neuron has the same RF structure with the frozen cue stimulus, the SI is 1; if the RF structure is uncorrelated with the original, the SI is 0. The SI of the responses to the control and the original stimuli measures the reproducibility of the RF. Of the neurons with spatially localized RFs, $53 \pm 5\%$ ($n = 51$) had a "reproducible" RF (the SI of the control dataset was significantly ($p < 0.01$) >0), and only these neurons were used in the following analysis. The SI value of the control dataset was used to normalize

the SI values of the frozen cue results (normalized SI [NSI]: NSI = SI/SI$_{control}$). The loss of NSI (1 − NSI) indicates the dependence of an RF to a particular sound localization cue.

The horizontal RFs of all the neurons with a reproducible RF are shown in Fig. 3k–o. Overall, spectrum freezing caused a drastic change to the RFs, leading to the largest loss of NSI, followed by ILD freezing, while ITD freezing did not cause a significant loss of NSI (Fig. 3p). These results highlight the importance of spectral cues for the calculation of auditory RFs in the SC.

**Heterogeneous use of spectral cues and ILDs across the SC.** The cue freezing experiment also revealed that neurons that rely on spectral cues and ILDs to compute RFs are not distributed evenly across the SC. Neurons with frontal RFs (in the anterior SC) have

a stronger dependence on spectral cues, while neurons with lateral RFs (in the posterior SC) have a stronger dependence on ILDs (Fig. 3q). When the neurons were divided into frontal and lateral groups at 45° azimuth, we found that the neurons in the frontal group relied almost completely on spectral cues to form their RFs, while those in the lateral group relied on both ILDs and spectral cues to form their RFs (Fig. 3r). As a consequence, when the stimulus spectra are frozen, most of the frontal RFs disappear and those that remain have a peak firing rate at ~65° azimuth (Fig. 3o, red dotted line), where the ILD is at its maximum (Supplementary Fig. 1c). This shows that spectral cues are a "necessary" component for computing an azimuthal topographic map in the SC.

**Monaural stimuli confirm the importance of spectral cues**. We were surprised to find that freezing the average ILD at zero did not have a more dramatic effect on the RFs of SC neurons. One reason for this could be that we only corrected the average ILD across the frequency range of 5–80 kHz. Because ILDs are a function of frequency, and can be as high as ±30 dB in a high-frequency range (see below), our zero average ILD stimuli may have had a remaining ILD in the high-frequency range that can be utilized by neurons (as suggested in a review study[19]). Therefore, we conducted two additional experiments to clarify the role of ILDs in RF formation. In the first experiment, we presented monaural stimuli, which entirely lack physiologically relevant ILDs and ipsilateral spectral cues; in the second experiment, we extended the ILD stimuli to include a range of ±40 dB ILD.

When we compared the RFs of neurons generated from binaural (control) and monaural stimuli with and without spectrum freezing, we found that monaural stimuli led to a loss of lateral (Fig. 4i, m), but not frontal RFs (Fig. 4d, i, m). In this experiment, we recorded from 869 neurons from five mice, and found that $59.5 \pm 1.6\%$ ($n = 554$) had a significant auditory response, $12.6 \pm 1.4\%$ ($n = 70$) of these had a localized RF, and $45 \pm 6\%$ ($n = 32$) of these had a reproducible RF. An example neuron with a frontal RF is shown in Fig. 4a–e. Note that spectrum freezing, but not monaural stimuli, led to a loss of SI. Figure 4f–j show a neuron that has an RF that is altered both by spectrum freezing and monaural stimuli. In this neuron, the response to the sound coming from the front was not changed during monaural stimuli (blue arrows), but the difference in the responses to ipsilateral and contralateral stimuli goes away during monaural stimuli (Fig. 4i, j, red arrows). This trend holds for every neuron with a localized RF (Fig. 4k–n), suggesting that the spectral cues are responsible for frontal RFs, and ILDs are responsible for lateral RFs. The overall NSI loss for each frozen cue was similar, but they act on different parts of the RFs; hence, freezing both cues leads to the complete loss of NSI (Fig. 4l–o).

We also used extended ILD stimuli to see if neurons are tuned to a specific value of the ILD or if the firing rate increases monotonically with the ILD. We kept the spectrum of the stimulus flat, but changed ILDs randomly across the range from −40 to +40 dB (see Methods). We recorded from 856 neurons from five mice, and found that $9.2 \pm 1.0\%$ ($n = 79$) of the neurons had significant and non-flat auditory responses across the range of ILDs tested (Fig. 4p). Only one neuron displayed a peaked tuning (a peak firing rate that appears anywhere between −40 and +40 dB, and is significantly ($p < 0.01$ after Bonferroni correction) higher than the firing rate at both ±40 dB). This means that the firing rate of most neurons increased monotonically as a function of the ILD (Fig. 4p), supporting the observations in Figs. 3o and 4l that spectral-cue freezing makes the RF of ILD-dependent neurons shift to where the ILD is maximized (at ~65° azimuth; red dotted

line in Fig. 3o). These results suggest that the neurons cannot form a topographic map by ILDs alone.

**Determining the spectral pattern associated with spatial RFs**. To identify the features in the HRTF that give rise to the spectral-cue-based frontal RFs and the ILD-based lateral RFs, we measured the STRFs of the neurons with localized RFs by stimulating each ear with uncorrelated dynamic random chord stimuli and calculating the spike-triggered average (STA)[20] from stimuli presented at each ear (Fig. 5a, see Methods for details). STRFs characterize the spectral structure and delay time of stimuli that correlate with an individual neuron's spikes. Figure 5 shows our results summarizing the recordings of 590 neurons from three mice, $55 \pm 2\%$ of which ($n = 323$) had significant responses to this stimulus, $25 \pm 2\%$ of these ($n = 80$) had a localized spatial RF, and $49 \pm 6\%$ of these neurons ($n = 39$) had a significant STRF (see Methods for details of the significance test). An example STRF for a neuron with a frontal RF is shown in Fig. 5b. This neuron's preferred stimulus is tones in the ~48–60 kHz frequency range (red pixels) without tones at ~40 and ~70 kHz (blue pixels), which are at 10–20 ms before the spikes (Fig. 5b). This neuron has similar frequency tuning properties between the ipsilateral (left) and contralateral (right) ears, which result in a positive binaural SI (BSI, calculated similarly to the SI in the cue freezing section). Figure 5c shows another neuron with a frontal RF, which has a similar STRF but exhibits significant frequency tuning only to sound in the contralateral ear, leading to a near-zero BSI. In contrast, a neuron with a lateral RF prefers frequency structures in the contralateral ear that are similar to the neurons with a frontal RF, but differs in that the preferred frequency structures between the ipsilateral and contralateral ears are anti-symmetric, resulting in a negative BSI (Fig. 5d).

To quantify the difference between the STRFs of neurons with frontal RFs and those with lateral RFs, we grouped them based on their RF azimuths (frontal <35°, $n = 19$; Fig. 5e, f and lateral ≥35°, $n = 21$; Fig. 5g, h; divided at 35° instead of 45° to include a similar number of neurons in each group) and determined the fraction of neurons that had a significant structure in each STRF pixel. Comparing Fig. 5e–h shows that while both frontal and lateral groups have positive STRF structures near 10–24 and 48–60 kHz ranges in the contralateral ear within 20 ms (bands indicated by magenta color), the frontal group has negative components in the 25–45 and 63–76 kHz ranges (Fig. 5f, green arrows in bands indicated by cyan color), which are not observed as much in the lateral group (Fig. 5h). Also, while the frontal group has positive STRF components in the 48–60 kHz range in the ipsilateral ear, the lateral group shows a negative structure in this frequency range (Fig. 5h, green arrow). The increased number of negative structures in the ipsilateral STRF is consistent with the BSIs of individual neurons that progressively decrease as a function of the RF azimuth (Fig. 5i).

The STRF properties of the frontal group (Fig. 5e, f) are similar to the spectral structure of the frontal HRTFs, which contains alternating amplified and attenuated spectral structures in the 10–80 kHz frequency range indicated by magenta and cyan colors (Fig. 5j, k). The within-ear spectral structure of the HRTFs may not be used by the lateral group because we find that their spiking depends less on a specific spectral structure (Fig. 5g, h), but depends more on the contrast between the two ears. Correspondingly, the contralateral ear has a larger gain than the ipsilateral ear in a broad range of frequency in the lateral azimuth where ILD is maximized (~65° in Fig. 5l). These results suggest that neurons with a frontal RF fire when contrast at different frequencies is presented into the same ear (i.e., spectral cues) and neurons with lateral RF fire when contrast between the different ears is

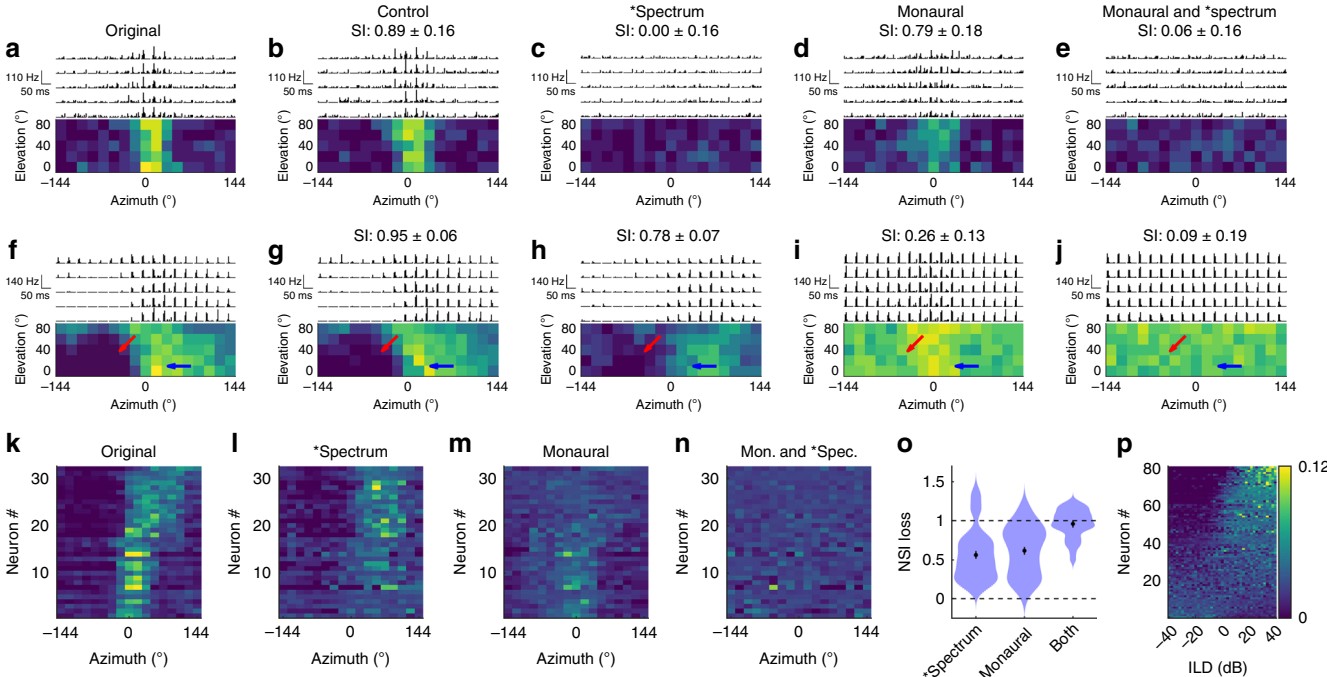

**Fig. 4 Structural changes of auditory RFs due to spectrum freezing and monaural stimuli. a–e** Response properties and RFs of a neuron in response to **a** binaural original and **b** control stimuli, **c** binaural stimuli with spectrum freezing, **d** monaural stimuli, and **e** monaural stimuli with spectrum freezing. This neuron loses its RF with spectrum freezing (**c**, **e**), but not with monaural stimuli (**d**). **f–j** An example of a neuron whose RF is comprised of spectral-cue and ILD components. Notice how the lateral part of the RF remains intact during spectrum freezing, suggesting that this part is based on ILDs. The frontal part of the RF remains intact with monaural stimuli, suggesting that this part is based on monaural spectral cues. With the monaural stimulus, the ipsilateral side has responses (red arrow) that are not seen in binaural stimuli (**f–h**). **k–n** Horizontal RFs of 32 neurons. For each neuron, the scale of the responses is normalized by dividing by the sum across all directions. The range of the color scale is from 0 to 0.3. Spectrum freezing results in distorting the frontal RFs and maintaining the lateral RFs (**l**). Monaural stimuli result in maintaining the frontal RFs, distorting the lateral RFs and reducing the suppression of the ipsilateral RFs (**m**). When both are implemented, all the RFs disappear (**n**). **o** Summary of the NSI loss to spectrum freezing, monaural stimulation, and both. Spectrum freezing and monaural stimuli have a similar level of NSI loss, but their roles are different. When both are implemented, the NSI is close to 1, meaning that the RFs are completely lost. Source data are provided as a Source Data file. **p** Responses of the ILD-sensitive SC neurons to an extended range of ILDs. The firing rates (FRs) are normalized by the sum across the full ILD range for each neuron. Neurons with non-flat responses to ILDs (a flat distribution fit gives a large $\chi^2$ value ($p < 0.01$)) are shown. For most neurons, the FR changes monotonically as a function of the ILD, and is not "tuned" to a specific ILD value.

presented (ILD). This supports our overall finding that each of these cues is important for the formation of an azimuthal topographic map.

## Discussion

In the present study, we found that: (1) mouse SC auditory neurons have spatially restricted RFs that form a smooth topographic map of azimuthal space; (2) ILDs and spectral cues, but not ITDs, are important for the generation of spatial RFs; (3) neural responses to sound from the frontal field depend on spectral cues, while responses to sound from the lateral field depend on ILDs and neurons can use both cues to tune its RF azimuth; and (4) the spectral tuning properties of neurons with a frontal RF match well with the spectral structure of the contralateral HRTFs of the frontal field. These findings are summarized in Fig. 6.

By recording from SC neurons in awake-behaving mice in response to VAS stimulation, we found that the mouse SC contains auditory neurons that form an azimuthal topographic map of sound, with the slope of azimuth approaching that of the visual map in the SC, albeit with a small but significant difference between the slopes of the visual and auditory maps. We do not yet know whether this difference is functionally important, but as suggested in a primate study[21], it is possible that downstream

processing can compensate for this small difference between these two maps.

Specifically, we found a strong correlation between the A–P position of the auditory neurons in the SC and their RF azimuths (Fig. 2e, $r = 0.70$). The correlation of the M–L neuron position and RF elevation was weaker (Supplementary Fig. 5, $r = 0.22$). This is consistent with a previous report that showed that mice can discriminate two sound sources better along the azimuthal axis than the elevation axis (azimuthal: $31 \pm 6°$; elevation: $80.7 \pm 1.7°$)[22].

An auditory topographic map along the azimuth has been reported in barn owls[7,8], ferrets[10], and guinea pigs[11,12], but whether one exists in rodents has been debated. Studies of auditory SC neurons employing anesthetized rats found either a weak correlation of the A–P positions and RF azimuths ($r = 0.41$, or $r^2 = 0.17$ in their report)[23], or a small slope between A–P positions and RF azimuths ($8.67° \text{ mm}^{-1}$ in fully adult rats; this value is small even considering the A–P extent of the rat SC is approximately twice as large as the mouse SC)[24]. Another study, which utilized anesthetized golden hamsters, found spatially restricted RFs only in the posterior SC[25]. We speculate that our use of awake-behaving animals is one reason that we found the spectral-cue components of RF and a topographic map in mice, in contrast to the previous studies. Other improvements in our experimental design include: the use of the VAS system instead of

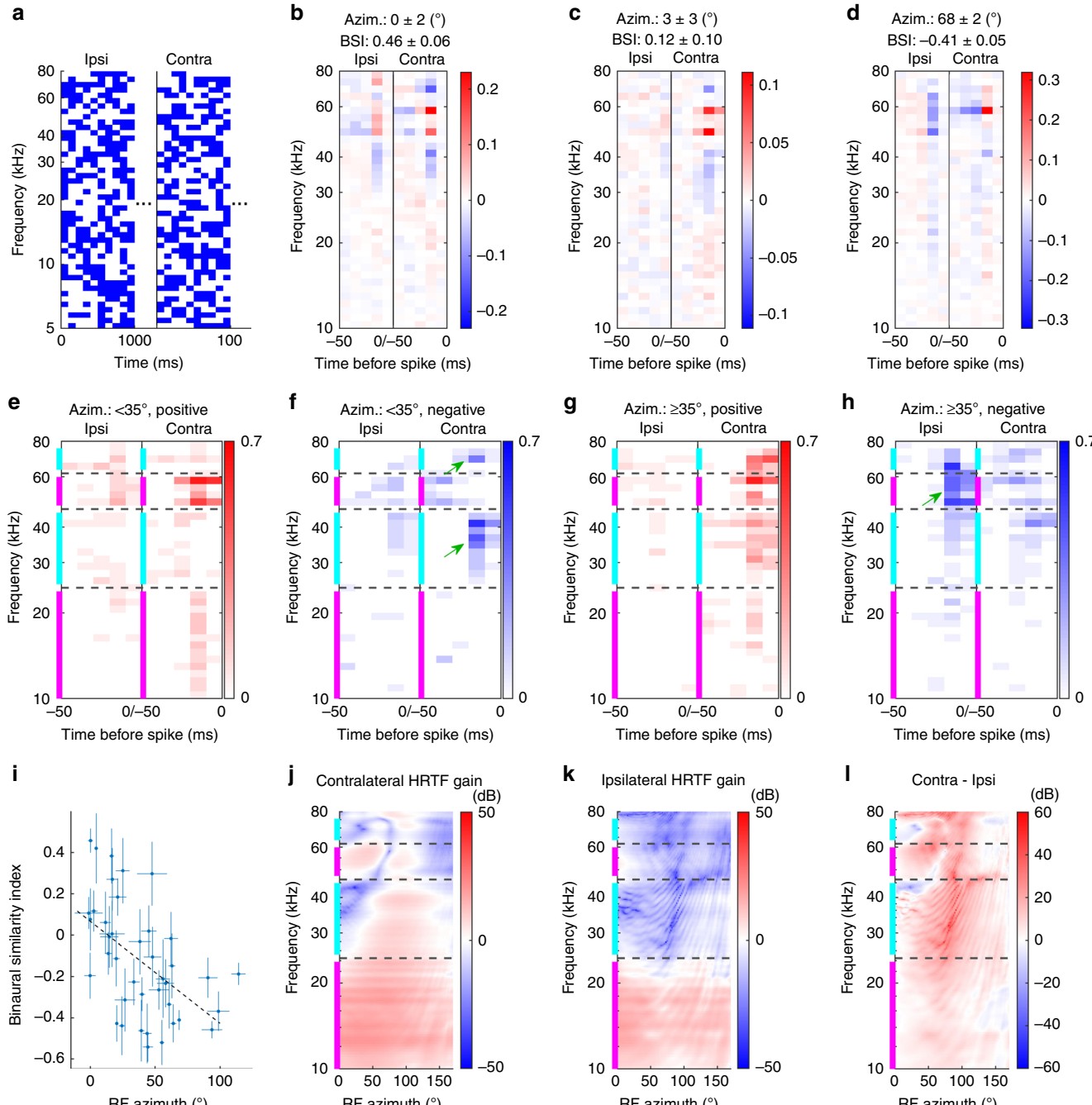

**Fig. 5 Spectro-temporal receptive fields (STRFs) of the neurons with spatially localized RFs. a** The first 100 ms of the dynamic random chord stimulus to the ipsilateral (ipsi; left) and contralateral (contra; right) ears. Blue pixels indicate the presence of a tone. **b–d** Example STRFs of neurons with frontal RFs (**b**, **c**) and a lateral RF (**d**). The pixel color indicates the deviation from the stimulus average at the corresponding frequency. (The stimulus average fluctuates around 0.5 because of the stimulus randomness.) The STRF structures of the two ears are symmetric (**b**), uncorrelated (**c**), and anti-symmetric (**d**), resulting in positive, near-zero, and negative values of the binaural similarity index (BSI; see Methods), respectively. **e**, **f** A summary of the positive (**e**) and negative (**f**) structures observed in the neurons with frontal RFs. The pixel color represents the fraction of neurons with a frontal RF that had a significant structure at the frequency/time before spike. Many frontal neurons exhibited positive structures in the 10–24 and 48–60 kHz ranges (magenta bands) and negative structures in the 25–45 and 63–76 kHz ranges (cyan bands) in the contralateral ear (green arrows in **f**). **g**, **h** Same as **e**, **f** for lateral neurons. In contrast to the frontal neurons, these neurons show positive structures in the contralateral cyan bands and negative structures in the ipsilateral upper magenta band (green arrow in **h**). **i** A scatter plot of the RF azimuth and the BSI. The errors in X-axis are derived from the Kent distribution fits; the errors in Y-axis are based on the Poisson statistics and arithmetic propagation (see Methods). These are anticorrelated ($r = -0.52$ ($p = 5e - 4$; using the "corrcoef" function in Matlab), slope: $-0.0049 \pm 0.0013°^{-1}$), indicating that lateral neurons prefer larger binaural contrast. **j–l** HRTFs measured in the horizontal plane for the contralateral (**j**) and ipsilateral (**k**) ears, and their difference (**l**). The values indicate the gain relative to a free-field measurement without the head. The spectral structures of the frontal part (small azimuth) of the contralateral ear (**j**) match well with the frequency tunings of the neurons in **e**, **f**.

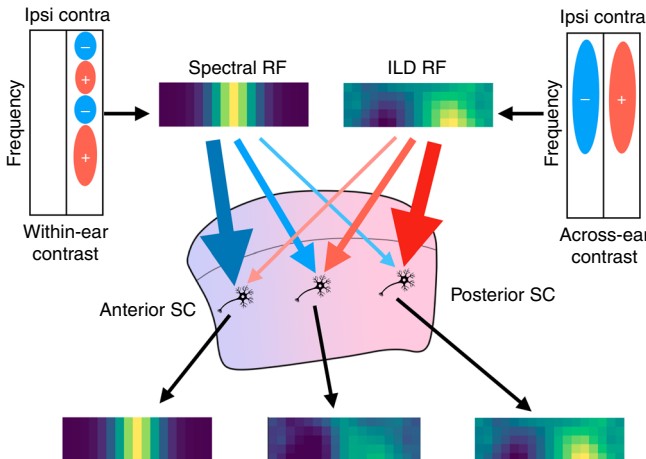

**Fig. 6 Combination of spectral cues and ILDs are used to form a topographic map of azimuthal auditory space.** Within-ear contrast of the input spectrum is used to provide an excitation to sound from the frontal field, while across-ear contrast (ILD) in a broad frequency range provides the excitation to sound from the lateral field (blue circles with − signs denote frequency bands that negatively influence the response; red circles with + signs denote frequency bands that positively influence the response). Neurons in the SC receive these inputs with different weights depending on their anteroposterior position. This progressive change of the weight creates the auditory topographic map of azimuthal space in the mouse SC.

a free-field speaker to deliver stimuli, which avoids acoustic interference of the sound with obstacles such as the recording rig; utilizing multichannel silicon probes to record from a large population of neurons in a single session increased the statistical power of our conclusions; thorough sampling of the auditory responses in two-dimensional spherical coordinates; RF parameter estimation through maximum-likelihood model fitting using appropriate error analysis with quasi-Poisson statistics; and finally, separating out the fast (<20 ms) component of responses of the neurons. Each of these elements contributed to the accurate estimation of the topographic map properties in the present study.

Our results indicate that 23% of the auditory-responsive SC neurons had a localized RF. This raises a question: what is the nature of the remaining 77% of auditory-responsive neurons? Are they simply neurons whose RF was below our significance threshold, or do they have unique functions such as modulating the response properties or calling attention to a threat? One possibility is that these neurons are used for visuoauditory integration and respond with a well-localized RF only when presented with simultaneous visual and auditory stimuli. It is known that visuoauditory integrative neurons exist in the deep SC[26], which may include some of these auditory neurons. Learning more about these neurons will give insights into mechanisms of sensory integration.

We found that spectral cues are required for frontal RFs and ILDs are required for lateral RFs, while ITDs play no role in spatial computation in the mouse SC. Our finding that spectral cues are necessary for generating spatially restricted RFs in the mouse SC should clarify their importance. Our work is consistent with that of Keating et al.[5], who showed that ferrets can be trained to perform azimuthal sound localization with one ear plugged, and that of Huang and May[27], who showed that cats exhibit poor head orientation toward a sound source when the mid-frequency spectral cues are eliminated from the stimulus. Moreover, our work extends previous studies that demonstrated that the RFs of

SC neurons in the guinea pig and the ferret are topographically organized along the azimuth when monaurally stimulated near their response threshold[13,14]. Interestingly, this map disappears when the stimulus sound is above the threshold values. In our experiments, we used stimuli that are well above the hearing threshold and also find that there is no azimuthal map when a mouse uses monaural hearing. In addition, we show that the spectral cues are "necessary" for the azimuthal topographic map even when ILDs are available.

We found that most ILD-dependent neurons have RFs in the lateral field, close to where the ILD is maximized (Figs. 3o and 4l). These neurons are not tuned to a specific level difference, but instead respond when level differences are greater than a given threshold (their response curve is a sigmoid-like monotonic function rather than a tuned, Gaussian-like non-monotonic response curve; Fig. 4p). This property is also found in the majority of neurons in the auditory nuclei that process ILDs, namely, the lateral superior olive and its afferent, the central nucleus of the inferior colliculus (IC)[28]. The SC receives indirect input from these nuclei through the external nucleus of the IC and the nucleus of the brachium of the IC[29]. Few non-monotonic neurons were found in the IC of bats[28] and primates[30] and the SC of cats[31] and primates[21]; consistent with these observations, we found only one non-monotonic neuron out of the 79 ILD-sensitive neurons. Therefore, when the stimulus spectra are frozen, most neurons change their RF azimuths to ~65° where the ILD is maximized (Figs. 3o, 4l, also see Fig. 5l and Supplementary Fig. 1c for the ILD as a function of the location). Although these neurons may potentially encode different sound source azimuths by using different activation thresholds (as suggested[28]), they still make an RF in the same position in the absence of spectral cues. All these data indicate that SC neurons use ILDs in combination with spectral cues to differentiate their RF positions (Fig. 6).

ITD freezing did not alter the RFs of the SC neurons. These results are consistent with previous reports, which concluded that ITDs do not contribute to the RF properties in ferrets[2] or marmosets[4], and the observation that the mouse has an underdeveloped medial superior olive[32], a nucleus that processes ITDs in other animals. This makes sense because the maximum ITD we measured for the mouse was only 29 μs (Supplementary Fig. 1d). Therefore, as predicted from the work of others, it is unlikely that ITDs contribute to sound localization in mice, as they are small-headed mammals[33]. There is a possibility that spectral cues are particularly more important for mice because they cannot utilize ITDs. Therefore, to examine the extension of our findings across species, the role of spectral cues in species that utilize ITDs for horizontal sound localization will need to be studied.

A potentially important difference between ILDs and spectral cues is that an appropriate interpretation of spectral cues requires knowledge of the original spectrum of the source sound. For a given stimulus, the ILD for a given source location is always the same (although it is a function of frequency). On the other hand, the spectra of the sound that arrives at the eardrums are influenced by both the spectra of the source sound and the HRTF, but the brain cannot tell them apart (this is exactly why VAS stimulation works). Therefore, if the spectrum of the source sound is an abnormal shape or restricted to a narrow frequency band, spectral cues will not be able to provide accurate information of the sound source location. Although humans can learn the abstract structure of sound rapidly through experience[34], whether such information can facilitate sound localization is not known. It will be interesting to test the tuning properties and relative importance of the spectral cues and ILDs using different types of auditory stimuli, including tonal sound and natural/naturalistic sound. It will also be interesting to test sound from different distances, a parameter that we did not examine in this study

because higher frequency sound is preferentially dissipated by traveling a long distance in the air.

To investigate what specific features of the sound localization cues give rise to the RFs of the neurons, we measured the binaural STRFs using the earphone setup that allowed us to stimulate each ear independently. Such an experiment has not been previously reported, and it revealed the frequency tuning properties as well as symmetry between the ipsilateral and contralateral ears, giving new insights into how different cues are used by SC neurons. Namely, we found that neurons with frontal RFs are positively influenced by sound with a general spectral pattern within the same ear, which coincides with that of the HRTFs for the sound from the front; neurons with lateral RFs are not as sensitive to this spectral pattern within an ear, but prefer to fire when a stimulus contrast is presented between the two ears.

The measured STRFs contain structures in a broad frequency range (10–76 kHz), with many peaks above ~50 kHz, suggesting that mice utilize these to localize sound. In the previous studies that aimed to measure sound localization acuity through behavior, stimuli with frequencies only up to 50 kHz were used[22,35,36]. These results will be worth revisiting using equipment capable of higher frequencies. In addition, ultrasonic vocalizations (USVs) that mice use to communicate are in this frequency range[37]. It will be intriguing to study the response properties of the SC neurons to USVs and their roles in communication.

To conclude, while ITDs and ILDs were thought to be the important sources for sound localization in the horizontal plane, we discovered that spectral cues are responsible for frontal RFs, and are necessary for tuning RF azimuths. Our results suggest that the formation of the topographic map of azimuthal auditory space in the mouse SC is accomplished by a gradual change of the relative weights of two simple RFs derived from spectral cues and ILDs (Fig. 6).

## Methods

**Ethics statement**. All procedures were performed in accordance with the University of California, Santa Cruz (UCSC) Institutional Animal Care and Use Committee.

**Measurement of the HRTF**. To measure the HRTF of the mouse, a pair of Golay codes[38] ($2^{16} = 65,536$ points for each) were played at a sampling rate of 500 kHz through a digital-to-analog converter (DAC; National Instruments (NIs), PCIe-6341), an amplifier (Tucker Davis Technology (TDT), ED1), and an open-field electrostatic speaker (ES1) toward a decapitated mouse head located 25 cm away. The response was recorded with a microphone (Bruel & Kjaer, 4138-L-006, frequency response: 6.5 Hz to 140 kHz) coupled with the back of the ear canal of the mouse, amplified (Bruel & Kjaer, 1708), low-pass filtered (Thorlabs, EF502, 100 kHz), and digitized (NI, PCIe-6341) with a sampling rate of 500 kHz. The head-related impulse response (HRIR) was reconstructed from the responses to the Golay codes[38]. The measurement was performed in an anechoic chamber (a cube with 60-cm-long sides) with egg-crate-shaped polyurethane foam attached to the ceiling, walls, and the floor of the chamber, as well as other surfaces such as the stage for the mouse head, the microphone and its cable, and the arm that holds the microphone.

Before measuring the HRTFs, the mouse was euthanized and then decapitated. The cochleae and the eardrums were removed from the ear canal and the microphone was coupled to the back of the ear canal using a ~1-cm-long coupling tube. In order to keep the tissue from drying and ensure a continuous seal between the ear canal and the microphone, a small amount of mineral oil was applied on the tissue around the ear canal. To measure the HRTF as a function of the incident angle, the mouse head was mounted on a stage that was coupled to a stepper motor that was driven by an Arduino board. The stage was rotated by the stepper motor and tilted by a hinge, without needing to detach the microphone (Supplementary Fig. 1a). We measured the HRTF for grid points of 10 elevations (0–90° with 10° steps) and 101 azimuths (0–180° with 1.8° steps), 1010 points in total, in the upper right quadrant of the mouse head. Note that the polar axis of the coordinate for this measurement is the rostral direction (Supplementary Fig. 7a). Before the microphone was detached, the HRTF of the setup with earphone (earphone HRTF, EHRTF) was also measured. This EHRTF was subtracted during the electrophysiological recording in order to avoid the additional frequency response due to the earphone setup. The same procedure was repeated for the left ear to achieve binaural HRTFs. Measuring the HRTFs from three female mouse heads

showed good reproducibility (the average difference was 5.3 ± 0.8 dB). We estimated the inconsistency of HRTFs between animals as an angular error (~±14°; see "Estimation of additional systematic errors of the RF parameters" in "Data Analysis") and incorporated this into our error measurements. The HRTFs were collected from female mice, but stimuli based on them were presented to both male and female mice. We did not see a difference in the topographic map parameters between male and female groups (Supplementary Fig. 8). In addition, even though their average body weights were significantly different (male: 28.2 ± 0.7 g ($n = 6$), female: 21.3 ± 1.5 g ($n = 6$), $p = 2 \times 10^{-3}$ (two-tailed $t$ test)), the ear sizes were not significantly different (the numbers represent [right ear long axis, right ear short axis, left ear long axis, left ear short axis]; male: [13.0 ± 0.3, 7.8 ± 0.2, 13.2 ± 0.2, 7.5 ± 0.1]; female: [12.6 ± 0.3, 7.9 ± 0.3, 12.6 ± 0.3, 7.9 ± 0.2]; $p = [0.39, 0.90, 0.12, 0.26]$). The LabView and Arduino programs written for HRTF measurements and the measured HRTFs are available online[39,40] (see Code availability and Data availability sections).

**VAS stimulation**. Using the measured HRTFs, we developed a VAS stimulation that creates stimulus sound with properties consistent with the sound coming from a specific source direction. This approach has been successfully used in humans[16], marmosets[4], guinea pigs[11], and ferrets[2,17]. The stimulus sound is first filtered by a zero-phase inverse filter of the ES1 speaker and the EHRTF, to flatten the frequency responses of the ES1 and the earphone setup. Next, the stimulus is filtered by a measured HRIR, including phase, to construct the sound property that is consistent with the incident angle for which the HRIR is measured. We only reproduced the frequency response in the range between 5 and 80 kHz because the ES1 speaker did not produce a sufficient amplitude outside this range. Within this range, the reconstructed sound reproduces the ITDs, ILDs, and spectral cues.

Sound is delivered to the ears through a DAC (NI, PCIe-6341), an amplifier (TDT, ED1), and closed-field electrostatic speakers (EC1) coupled with a small plastic horn. The tip of the horn was oriented toward the ear canal (~40° elevation, ~110° azimuth) and placed ~1 cm from it. With this angle, the EHRTF does not contain a strong notch and thus cancellation by an inverse filter was easy.

*Full-field VAS stimulus*: The full-field stimulus used grid points of five elevations (0–80° with 20° steps) and 17 azimuths (−144° to 144° with 18° steps), totaling 85 points in the two-dimensional directional field (Supplementary Fig. 7a). We measured the HRTFs only in the upper right quadrant because of the limitation of the measurement stage. In order to construct the HRTFs in the upper left quadrant, we copied the transfer function of the opposite ear, flipping left and right. This is done after confirming that the left ear HRTF and right ear HRTF are similar in the horizontal plane (left–right symmetry of the ear shapes). The measured ILDs and ITDs for the upper right quadrant are plotted in Supplementary Fig. 1c, d. The baseline stimulus pattern was 100-ms white noise with linear tapering windows in the first and last 5 ms. The stimulus was presented every 2 s and repeated 30 times per direction (total duration was 85 min). The stimulus intensity was 50 dB SPL. We generated a new pattern of white noise at every point of space and on every trial (the pattern is not "frozen"). Example stimuli are available in our Figshare repository[41] (see Data availability section).

Because we used an open-air type earphone, some sound from the earphone is also detected by the contralateral ear. We measured this cross-talk amplitude to be ~−10 dB at 5 kHz and <−30 dB at 30 kHz and above, relative to the stimulated ear.

*Freezing each sound localization cue*: Freezing a specific sound localization cue was achieved by fixing the cue for all incident sound directions while leaving the other cues to change their properties naturally as a function of sound direction. Stimuli with frozen cues were randomly interleaved so that the long-term change of the recording condition does not influence the differences of the RFs. For this experiment, we repeated the stimuli 20 times per direction instead of 30 times to shorten the total duration (with 5 conditions, 85 directions, and 2 s intervals, the total duration was 4.7 h). To freeze the ITDs, we first measured the peak timings of the HRIR for each location in each ear and shifted the left HRIR to set the timing difference to zero. To freeze the average ILDs, we calculated the average amplitudes between 5 and 80 kHz in left and right HRTFs and adjusted their amplitudes to be the average of the left and right HRTFs so that they have equal overall sound levels across this range of frequency. To freeze the spectral cues, we replaced the HRIR with a single sample impulse to flatten the spectrum while the overall sound level and timing were kept consistent with the original sound. To create monaural stimuli, we simply sent no signal to the ipsilateral (left) speaker. This effectively kept the ILD at a high value for all sound directions.

Note that freezing a sound localization cue does not mean erasing the cue. The frozen cue is still present at every location and may give information that contradicts other cues. For example, zero ILDs are consistent with a sound that comes from the midline, and some neurons may respond in a non-natural manner when this is combined with spectral cues that vary in the virtual space.

*Extended ILD stimulus*: To test whether the neurons are tuned to larger ILD values, we used the same 100-ms white noise bursts with a flat spectrum but changed the ILD randomly between ±40 with 2 dB increments. The stimulus for each ILD value was repeated 30 times.

*Dynamic random chord stimulus*: To measure spectral tuning properties of the neurons with localized RFs, we used dynamic random chord stimuli[20] and calculated the STA of the stimuli for each neuron. The stimuli consist of 48 tones ranging from 5 to 80 kHz (12 tones per octave). Each pattern was 10 ms long with

3 ms linear tapering at the beginning and the end (i.e., plateau for 4 ms). We used a short duration of the patterns to detect fast (<20 ms) responses of the SC neurons. In each pattern, the tones were randomly set to either ON or OFF with a probability of 0.5. The total number of tones per pattern was not fixed. The amplitude of each tone was fixed and set to be 50 dB SPL when averaged over time. One presentation of the stimuli was 2-min long (12,000 patterns) and this presentation with the same set of patterns was repeated 20 times to produce a 40-min-long stimulus. Tones from the left and right speakers were not correlated with each other in order to measure the tuning to contrast between the ears. We did not use a specific HRTF for this experiment, but simply canceled the earphone HRTF so that the stimulus sound was filtered to have a flat frequency response near the eardrums.

**Animal preparation for electrophysiology.** We used 2–5-month-old CBA/CaJ (The Jackson Laboratory, 000654) mice of each sex. The mice were kept in cages at 20–23 °C temperature, 30–70% humidity level, and a 12-h light/dark cycle. Mice were co-housed whenever possible and sufficient food and water were provided all the time. One day before the recording, we implanted a custom-made titanium head plate on a mouse's skull, which allowed us to fix the mouse's head to the recording rig without touching the ears. On the day of the recording, the mouse was anesthetized with isoflurane (3% induction, 1.5–2% maintenance; in 100% oxygen) and a craniotomy was made (~1.5–2 mm diameter) in the left hemisphere above the SC (0.6 mm lateral from the midline, on the lambdoid suture). We did not use ear bars to attach a head plate or to perform craniotomy in order to avoid damaging the ears. The mouse was given >1 h to recover from the anesthesia before recording. The incision was covered with 2% low-melting-point agarose in saline and a layer of mineral oil on top of it to keep the brain from drying. A 256-channel silicon probe (provided by Prof. Masmanidis[42]) was inserted through the cortex into the SC with its four shanks aligned along the A–P axis (Fig. 1b; the electrodes were facing toward the lateral direction). When the shank enters the superficial SC, the positions of multi-unit visual RFs on each shank are recorded. We then lower the probe until the strong multi-unit visual responses disappear. As a consequence, the top of the active area of the most superficial shank is located at ~300 μm from the surface of the SC. Before fixing the probe at the final location, the probe was overshot by ~120 μm and rewound in order to reduce probe migration during the recording. Recordings were started 20–30 min after inserting the probe. The mice were euthanized after the recording session.

During the recording, the mouse was allowed to run freely on a cylindrical treadmill made of polystyrene foam (Fig. 1a). The surface of the cylinder was covered with self-adherent wrap (CVS pharmacy) to reduce locomotion noise. The movement of the cylinder was recorded by a shaft encoder (US Digital, H5-100-NE-S).

In five of ten experiments in the "Mice have spatially localized auditory RFs that are topographically organized" section, we inserted the probe three times per mouse in order to span the recordings in different M–L positions to measure the elevation tuning. Each penetration was separated by 400 μm along the M–L axis. The order of penetration in the M–L axis was randomized for each experiment to avoid a correlation between a degradation in recording quality with the probe location. All the other mice had only one probe insertion.

**Data analysis.** *Blind analysis*: We performed a blind analysis of our data, which is effective for reducing a false-positive reporting of results[43,44]. First, we looked for specific features using half of the neurons randomly subsampled from each experiment (exploratory dataset). After deciding which parameters to look at, we performed the same analysis on the hidden half of the data (blinded dataset) to confirm if the results agreed with the exploratory dataset. All of the results reported in the present study passed a significance test both in the exploratory dataset and the blinded dataset unless noted otherwise.

*Spike sorting*: We used custom-designed software for spike sorting. Raw analog signals were high-pass filtered (cutoff ~313 Hz) and thresholded for spike detection. Detected spikes were clustered using the principal component analysis to its waveforms on the detected electrode and its surrounding electrodes. A mixture of Gaussians model was used for cluster identification. Clusters with many spikes in the refractory period and duplicated clusters for the same neuron (determined by isolation distance, L-ratio, cross-correlation analysis, and similarity of the waveform shapes) were excluded from the analysis. Further details are available in published articles[45,46].

**Error estimation for categorical numbers.** When we evaluate categorical populations such as a fraction of neurons with a fast response, we assumed a binomial distribution. If the probability that a neuron is in a category is $p$, its error is estimated by $\sqrt{p(1-p)/N}$, where $N$ is the total number of neurons.

**Estimation of the A–P locations of the neurons.** We estimated the relative position of the silicon probes in multiple experiments using the visual RF positions measured in the superficial SC. When we inserted a silicon probe, we measured the positions of the visual RFs on each shank in the superficial SC using multi-unit activity. We extrapolated the visual RFs to find an A–P position where the visual RF azimuth was 0°, and defined this point as the zero of the A–P position. To

estimate the A–P position of the auditory neurons we assumed: (1) the most superficial electrode of the silicon probe was at 300 μm in depth; and (2) the insertion angle of the probe into the SC was 25°. (1) is justified by consistently stopping penetration at the position where visual responses predominantly disappear from all of the electrodes; and (2) is justified by the post-recording observations of the insertion angle (Fig. 1c). The position of a neuron relative to the probe was determined by a two-dimensional Gaussian fit to its spike amplitude across multiple electrodes[46]. We only analyzed neurons with a positive A–P position in order not to include neurons outside the SC.

**Significance test for the auditory responses.** We used quasi-Poisson statistics for significance tests of the auditory responses of individual neurons[47]. Simple Poisson statistics was not sufficient because the post-stimulus firing rate typically had a larger trial-by-trial variance than that expected from Poisson statistics (overdispersion) due to factors such as bursting of the neural activity and/or locomotion/movement of the animal. These additional fluctuations can cause increased false positives. (The overdispersion parameter was estimated by the variance of the spike count divided by the mean, which should be 1 if the neuron is Poissonian. The average of the overdispersion parameters of all neurons was 5.61 ± 0.10. These parameters during stationary and running periods were 4.99 ± 0.08 and 5.11 ± 0.10, respectively, indicating that locomotion is one of the sources, but not a sole source of overdispersion.)

To determine the significance of the response, we first estimated an overdispersion parameter and considered a response to be significant if the $p$ value of the neuron's spike count is below 0.001 ($p = 1 - CDF(N)$, where CDF is a cumulative distribution function of the quasi-Poisson distribution and $N$ is the spike count of the neuron). We have chosen to use quasi-Poisson statistics over a negative binomial distribution (another distribution that permits overdispersion) because of its simplicity. We do not expect a large difference in results because of the choice between these two distributions.

**Function fit to estimate the azimuth and elevation of the RFs of the neurons.** We used a maximum-likelihood fit of the Kent distribution[18] to estimate the azimuth, the elevation, and the radius of an RF. The Kent distribution is a spatially localized distribution in a two-dimensional directional space. We have chosen the Kent distribution over the two-dimensional Gaussian distribution because we consider a large range of angles in the directional space in which the nonlinearity of the coordinate system should be taken into account.

The equation of the Kent distribution is given by:

$$f(\mathbf{x}) = \frac{1}{\exp(\kappa)} \exp\{\kappa \boldsymbol{\gamma}_1 \cdot \mathbf{x} + \beta[(\boldsymbol{\gamma}_2 \cdot \mathbf{x})^2 - (\boldsymbol{\gamma}_3 \cdot \mathbf{x})^2]\},$$

where $\mathbf{x}$ is a three-dimensional unit vector, $\kappa$ is a concentration parameter that represents the size of the RF ($\kappa > 0$), $\beta$ is an ellipticity parameter ($0 < 2\beta < \kappa$), the vector $\boldsymbol{\gamma}_1$ is the mean direction of the RF, and vectors $\boldsymbol{\gamma}_2$ and $\boldsymbol{\gamma}_3$ are the major and minor axes of the Kent distribution. These three unit vectors are orthogonal to each other. This function is normalized to 1 when $\mathbf{x} = \boldsymbol{\gamma}_1$. At the limit of large $\kappa$, this distribution becomes asymptotically close to a two-dimensional Gaussian. At each point of the directional field, the likelihood value was calculated based on quasi-Poisson statistics[47]. The error of each parameter was estimated from the Hessian matrix of the likelihood function.

We used the front-Z coordinate system (Supplementary Fig. 7a) for the HRTF measurement and the stimulus presentation, but used the top-Z coordinate system (Supplementary Fig. 7b) for the Kent distribution fitting. We had to use the front-Z coordinate system for the HRFT measurement because of the restriction of our measurement stage (Supplementary Fig. 1a). However, the front-Z coordinate system has a discontinuity of the azimuth across the midline that gives a problem in fitting and interpretation of the data near the midline. We avoided this issue by switching to the top-Z coordinate system. Because the Kent distribution in vector format is independent of the coordinate system, likelihood values and parameters in the vector format are not affected by this change when the fit is successful. To avoid an area where two coordinate systems are largely different, we only used neurons with their elevation smaller than 30° for azimuthal topography (Fig. 2e). To achieve stable fits, we restricted $\beta$ to satisfy $4\beta < \kappa$, set the azimuthal range to be from −144° to +144°, and set the elevation range to be from 0° to 90°.

We used the following equation to define the radius $\rho$ from the concentration parameter $\kappa$:

$$\rho = \cos^{-1}\left(1 - \frac{1}{2\kappa}\right).$$

With this definition, the value of the distribution becomes $\exp\left(-\frac{1}{2}\right)$ of the peak value when $|\mathbf{x} \cdot \boldsymbol{\gamma}_1| = \cos(\rho)$ and $\beta$ is negligible, a behavior similar to a two-dimensional Gaussian.

**Estimation of additional systematic errors of the RF parameters.** We estimated additional systematic errors that may affect the RF parameters. First, the mouse was alert and freely running on a cylindrical treadmill during the neural recording session. Locomotion modulates the functional properties of the visually responsive neurons[46,48,49] and the auditory cortical neurons[50,51]. To determine the effects of

locomotion on the auditory SC neurons, we separated the session into segments when the mouse was running (speed >1 cm s$^{-1}$) and when it was stationary. The mouse was running, on average, 30% of the time (Fig. S6a). During locomotion, the spontaneous activity increased and the auditory responses decreased both in the fast and slow timescales (Supplementary Fig. 6b–g). However, it did not change the individual RF structures or the map of auditory space (Supplementary Fig. 6h, i). Therefore, we did not consider this as a source of systematic error. Second, the mouse was moving its eyes during the experiment. In primates, eye movement influences the auditory RF of the SC neurons[52]. However, this property has not been characterized in mice and the range of the mouse eye movement is small. The standard deviation (SD) is ~2–3° and the saccade amplitude is <10° along the horizontal axis and the movement along the vertical axis is even smaller[48,53]. Therefore, it is unlikely that this is a major source of systematic errors. Based on the SD value, we take ±3° to be the estimated systematic error. Third, the HRTFs can be modulated by the movement of the mouse pinnae. To estimate the effect of the pinna movement on the HRTF, we measured the amplitude of the ear movement using an infrared video camera (Basler acA640-120 um) while the mouse listened to the auditory stimuli. We quantified the pinna angle in 100 randomly selected images of the video recording. Overall, the mouse moved the pinna by 13° (SD). Fourth, our VAS stimuli might not be perfect because we used a single set of HRTFs for all the mice instead of measuring them individually. In order to quantify this effect, we measured HRTFs from three mice and compared the differences (Supplementary Fig. 1b). The overall difference (RMS) was 5.3 ± 0.8 dB, and this corresponds to an angular shift of ±14°.

To estimate the topographic map parameters, taking into account both the systematic and statistical errors, the systematic errors noted above were added in quadrature to give ±19°. This is larger than the average statistical errors associated with the individual neurons (±5.8°). However, a linear fit to the topographic map along the azimuthal axis (Fig. 2e) resulted in a large $\chi^2$ per degree of freedom (24.1), which is in the regime of either underfitting or underestimation of the error. In order to estimate the slope and offset parameters, we added, in quadrature, the estimated systematic error of ± 19° to each data point. The $\chi^2$ per degree of freedom, based on a linear fit, is now 0.69, indicating that the systematic errors are a reasonable addition to the statistical errors.

**Measuring the effect of freezing each sound localization cue on the RFs**. We used cosine similarity to compare the similarity of the auditory RFs before and after freezing each cue. We considered the auditory responses at 85 virtual source locations as an 85-dimensional vector and normalized it so that it has a unit length and zero mean. Then we took an inner product of the responses to the original stimulus and the additional stimuli (control, frozen ITD, frozen ILD, frozen spectrum) to calculate the cosine SIs. The statistical errors of the spike count at each virtual source location (estimated by Poisson statistics) were propagated to the final error of the SIs. The SIs of the cue freezing experiments are further normalized by dividing by the SI of the control stimuli (NSI). The same cosine SI was also used to measure the similarity between the ipsilateral and contralateral STRFs (BSI).

**Estimation of STRFs**. We estimated the STRFs of neurons using the STA calculated from the spiking response to the dynamic random chord stimulus described above. First, the spikes of each neuron is discretized in time with 10 ms bins (the same as the stimulus segment size) and the average of the stimulus patterns that preceded each spike was calculated. The stimulus was considered as 1 if there is a tone in the frequency and 0 otherwise. The significance tests for the STRFs were performed by examining whether the difference between the mean of the stimulus and the STA is significant. Namely, the $p$ value was calculated for each pixel using a cumulative distribution function of the binomial distribution (in Matlab, this is using a function $binocdf(S*N, N, M)$, where $S$ is the calculated STA, $N$ is the number of spikes, and $M$ is the mean value of the stimulus over time). The mean value of the stimulus is close to 0.5, but not exactly because of the natural fluctuation of the stimulus. This value was evaluated for each frequency separately. If the $p$ value is more than $1 - 0.001/N_{pixels}/2$, the pixel is considered to have a significantly positive structure; if it is $<0.001/N_{pixels}/2$, the pixel is considered to have a significantly negative structure. $N_{pixels}$ is the number of pixels that we considered for a significant structure (Bonferroni correction factor), and in this experiment, it was 960 (10 time bins, 48 frequencies, and 2 ears). The extra factor of 2 is necessary because this is a two-sided test. This affirms $p < 0.001$ threshold level for the entire STA for the neuron. Neurons with <20 spikes during the stimulus were not analyzed.

**Reporting summary**. Further information on research design is available in the Nature Research Reporting Summary linked to this article.

## Data availability

The measured HRTFs for three CBA/CaJ mice and the explanation for their use are available in the figshare website with the identifiers https://doi.org/10.6084/m9.figshare.11690577 and https://doi.org/10.6084/m9.figshare.11691615. The example virtual auditory stimuli are also available in the figshare website with the identifier https://doi.org/10.6084/m9.figshare.11691717. The source data underlying Fig. 3p, r and

4o are provided as a Source Data file. The other data that support the findings of this study are available from the corresponding author upon reasonable request.

## Code availability

Custom LabVIEW and Arduino programs that are used for the HRTF measurement and analysis code with sample dataset are available in the corresponding author's Github page (https://github.com/shixnya/SoundLocalizationResearch). The other codes for data analysis of this study is available from the corresponding author upon reasonable request.

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

## Acknowledgements

This work was supported by Brain Research Seed Funding provided by UCSC, the National Institutes of Health (Grant NEI R21EY026758 to D.A.F. and A.M.L.), a Whitehall Foundation grant to D.A.F., and a donation from John Chen to A.M.L. We thank Sotiris Masmanidis for providing us with the silicon probes; Ben Abrams, Director of the UCSC Life Science Microscopy Center, for training and assistance with imaging; J. Gerard G. Borst, Jena Yamada, Sadaf Abed, and Youngmin Yu for helpful comments on the manuscript.

## Author contributions

S.I., D.A.F., and A.M.L. conceived the project. S.I. designed the experiments. S.I. and Y.S. acquired the data. S.I. and Y.S. analyzed the data. S.I., D.A.F., and A.M.L interpreted the results and wrote the manuscript.

## Competing interests

The authors declare no competing interests.
