## [Peer Review File · Nature Communications]

Reviewers' Comments:

Reviewer #1:

Remarks to the Author:

This paper offers an interesting and technically sophisticated experimentation that generates data from a large number of cells and from a wide range of positions in space, and with a wide variety of stimulus manipulations that should offer enormous insights into the underlying mechanisms that give rise to the topographic map selectivity.

However, what is unfortunate is that the analysis feels highly incomplete. So while there are some interesting findings, for example, the finding that the topographic SC map is confirmed, and the dependence on ITD and ILD is further clarified. However, what is lacking is a more detailed consideration of the "spectral cue" and, specifically, the role of frequency tuning and inputs in determining the RFs of the cells. The authors are well aware that frequency tuning makes a difference to the way a cell responds to the ILD,s and to the spectral cues. Yet, there is no consideration of this parameter at all. For instance, it is unclear whether nasal and temporal RFs are tuned in different ways, or mixed tuning. In fact, usually spectral cues contribute to localization through the frequency tuning of the cells, and since this is important, it is unclear how the spectral cues contribute heavily to the nasal RFs. Is it simply the lack of ILD sensitivity in the nasal RFs that uncovers them? Or is the nasal RF a sculpted spectral sensitivity that is tuned to the HRTF from central locations? Curiously, ODD sigmoidal ILD dependence, and the EVEN monaural sensitivity near the nasal locations (Figs 3m,o) give rise nicely to the total RF maps (e.g., Fig3k)!

Because of the lack of any information and analysis into the frequency selectivity of the various neuronal populations, one is left with a really simple "curve fit" of a model which is not really a model in the true sense of the word, but simply a (as is well described) a simplistic description of a gaussian RF shifting left and tight as one changes the weight of the ODD to EVEN terms. One learns nothing from such a model as to how the RF's arise in the first place. I highly recommend that the authors consider the contribution of the spectral tuning to the RF formation in different cells, and not just lump it all as the "spectral cue". As the authors point out, the spectrum contains very large implicit ILD's that are highly specific to certain frequency channels and these cannot be simply removed by equalizing the average level (as is well explained in the control experiments).

Finally, I strongly recommend changing the terms "nasal" and "temporal" to perhaps "central" and "lateral". The word "temporal" really throws off the reader into time-domain issues which are clearly not at all what the authors want.

Reviewer #2:

Remarks to the Author:

To date, most of the studies on the neural mechanisms of sound localization have largely focused on two major features: interaural timing differences (ITDs) and interaural level differences (LTDs). Here, the authors make an important contribution to the field by systematically assessing a third, largely understudied contributor to sound localization, spectral cues.

The findings are novel and interesting, the work is experimentally well conducted and of relatively board relevance for the neuroscience community. Overall, the work seems a good fit for the journal and we support its publication, however we do have a few minor comments and we would be grateful if the authors could address them in their response.

Head related transfer function

- It would be interesting to see a graph of the difference between the contra and ipsi HRTFs (fig S1c minus fig S1d). This would give the ILD across azimuths and frequencies.
- Defining the head related transfer function is not a minor feat, the authors have done an excellent work that will greatly benefit the community. With that regard, we wonder whether the obtained HRTF could be made more easily accessible to the community to simplify the work of other groups that might want to implement a similar virtual auditory space. I guess one way would be to publish the set of coefficients for an FIR filter that replicates the transfer function.

Use of spectral cues and ILDs across the SC for auditory RFs

- The paper states that auditory RFs are formed using ILDs and spectral cues. However, it seems that in the absence of both, some neurons still maintain an auditory RF (neurons 30 to 50 in figure 2k-o). Could the authors provide an additional graphic to figure 2k-o, showing the RFs of neurons with reproducible RF under simultaneous freezing of both spectrum and ILD (and eventually ITD)? This will constitute a negative control, and these neurons should lose completely their auditory RF. If that's not the case, it would point towards another error of the virtual auditory space stimulation, in addition to the estimation of the RF size and location, and should be commented also in the methods section ("Estimation of additional systematic errors of the RF parameters"). We do appreciate that figure 3 partially addresses the above concern by saturating the ILD to a maximum.

Modelling

- Figure 4b, it should be "fraction of neurons" instead of "# of neurons"
- It is unclear how to simulate a monaural experiment with the model. The sentence "assuming that monaural stimuli have a large ILD (i.e. $ILD-RF = 1$) at every virtual location" is unclear. Practically, what changes occur to the 4 model parameters, w_1 , w_2 , k and b to simulate monaural stimulation experiments? Are these parameters unchanged and only the function $F_{ild}(x,0)$ set to a high constant?

Discussion

- As suggested in the discussion on the ILD-dependent neurons, this property is also found in the lateral superior olive and the central nucleus of the inferior colliculus. Could you state whether there are afferent or efferent connections known between these nuclei and the SC? Could it be possible then that collicular neurons inherit Temporal ILD RF component from these nuclei?
- I'd like to further a bit the utility of the model, "useful for improving our understanding of the system and making testable predictions". It is clear that this model help understanding the combination of the two RF components for auditory collicular neurons. However, a testable prediction resulting from this model is missing. For example, the question could be raised of the origin of these two RF components. Because these two components are orthogonal, it could be that they are computed in separate afferent brain area that converge onto these neurons. Or one component could be computed locally in the SC.
- It is also missing where and how could the spectrum RF be computed. Studying the frequency tuning curve of auditory collicular neurons could help solving these question.

Reviewer #3:

Remarks to the Author:

The authors investigate the contribution of different sound localization cues to the auditory receptive field structure of neurons in the awake, mouse superior colliculus. Their major findings are:

- 1). The authors found a topographic representation of sound location in a subset of their recorded neurons along the anterior-posterior axis of the superior colliculus.
- 2). The receptive fields to these neurons relied on spectral cues and inter-aural level differences (ILD)

but not inter-aural timing differences (ITD).

3). Furthermore the spectral cues contributed more to receptive fields concerning the nasal aspect of auditory space while ILD cues contributed more to receptive fields concerning the temporal aspect of auditory space.

The main topic of this paper is likely to be interesting to multiple fields, especially - sound localization, multisensory integration, and the burgeoning field investigating of the role the superior colliculus in cognitive functions. It is of particular interest because the authors show a role for spectral cues in azimuthal localization where they are often ignored (most research focuses on ITD and ILD of azimuthal localization). Overall, I think the evidence presented in the figures supports their major conclusions. However, I found the presentation of the data and the details about the methods concerning, what the experimenters did to collect/analyze the data is particularly opaque. I don't think the model presented in the paper adds anything substantial. I also have some minor concerns about the interpretation of the data.

General Comments:

- 1). There should be more details about the stimulus presentation – specifically, how many times were each stimuli presented at each virtual location? Were they presented in sequence? Were the frozen experiments blocked or randomly interleaved? How much time was there between sound presentations?
- 2). It is difficult to get a feel for the variance of the RF structure across the neurons. In the population plots you can get a sense but it is not explicitly stated what the error bar means (also see below for more specific comments on those plots).
- 3). Freezing of specific cues: I think this was a clever manipulation. However, the way it's discussed in the paper makes it seem like the cue vanishes and there is no more information carried via that cue. However, isn't that not the case? For instance, if you set the timing difference to zero doesn't the ITD then signal a sound source near 0 degrees. So when one cue is frozen, the sound now carries conflicting information about sound source. I don't think this is particularly alarming issue but might warrant some attention in the discussion. It'd be interesting if the frozen experiments were blocked though – since the SC tends to weight salient/reliable stimuli more strongly than irrelevant/unreliable stimuli. If a cue no longer provides information for many trials in a row would there be plasticity in the SC to stop responding to that cue?

Section Specific Comments:

Introduction:

- 1). It seems the way the duplex theory is presented as the end-all-be-all is somewhat strawman-y. Much research over the years has called into question its strict interpretation (e.g. ref #3) – so there have been “questions about the validity of the duplex theory of sound localization” for some time now. A slight reframing with a weaker interpretation of the duplex theory might be more accurate
- 2). “The SC is an ideal brain area to study sound localization because it contains spatially tuned auditory neurons and a topographic map of auditory space.” Although in some species there is clearly an auditory map of space in the SC, in others it's quite controversial. Since this paper is claiming to be the first report of a topographic auditory map in the SC of mouse – I'm not sure saying that the SC contains a map here is ideal. Perhaps the qualifier “in some species” should be added or some combination with the next sentence would help.
- 3). “demonstrate that the mouse is a useful model to study the mechanisms of auditory processing” I'm not sure this was 1) ever in question or 2) how this paper explicitly shows how is it a useful model. It certainly does not seem to be the most relevant model to humans if that is how the authors

define useful.

Methods:

Measurement of the HRTF:

1). You measured from 3 decapitated heads – were both male and female heads included? Since the recordings took place in both male and female mice could there be a larger bias for a particular sex? For instance if male mouse heads were used and they were on average larger than female heads (I'm not sure if there are large differences between male mice and female mice but I know it can be the case for rats).

Animal preparation for electrophysiology:

1). Were ear bars used in the craniotomy surgery and if so does this cause any swelling in the ear canal/pinnae that could also add to a systematic error?
2). "We then lower the probe until the visual response disappear" 1. How were the visual response probed? 2. There are visual responses in the deep layers of the SC so why would the visual response disappear?
3). How did the authors ensure the mouse was awake? Do the mice fall asleep occasionally and if so did that change the responses of the SC?

Significance test for the auditory response:

1). Are the data not overdispersed when the animal is still (it mentions that a possible cause of overdispersion is locomotion)?
2). I think this section could use some fleshing out – what is the justification for quasi-Poisson (as opposed to negative binomial – e.g. ref #34). What explicitly was done?

Function Fit to estimate the azimuth and elevation of the RFs of the neurons:

1). What is the justification for using the Kent distribution as opposed to another distribution (like a 2 dimensional Gaussian)?
2). Is the Kent distribution able to capture monotonic and non-monotonic RF structures equally well? This seems important – especially for the ILD analyses in the later figures.

Estimation of additional systematic errors of the RF parameters:

1). "eye movements modulate the map of auditory space to keep the visual and auditory maps aligned" This reference is from rhesus macaques where the existence of an auditory map in the SC is still debated. Additionally, the movement of the auditory RF is half of what it should be if they were to keep the auditory and visual RFs aligned – so the eye movements adjust the RFs but not to complete alignment.
2). Is there any pinna movement associated with sound presentation? Anticipation of sound? Were the left and right pinna moved independently or were they moved the same? I'm not sure looking at 100 random frames is the way to control for sound correlated pinna movements – though it seems fine for getting a range of pinna movements.

Results:

Mouse SC neurons have spatially localized auditory RFs that are topographically organized and associated figures:

Text:

1). "The neurons had a variety of temporal response patterns, with their peak response time having a bimodal distribution" I found this sentence a bit confusing – I was unsure if the bimodal distribution was talking about each individual unit having a bimodal response or the population. Adding the word

population may clarify – e.g. with the population peak response time having a bimodal population.

2). I'm not entirely sure where the error is coming from in probing things like # of responsive neurons or # of neurons with a particular response latency? E.g. 77.5 +/- 0.6% of neurons had a peak response faster than 20 ms – where is the 0.6 coming from? Is this due to something intrinsic the quasi-Poisson statistics? Was this due to the blind analysis? Is that averaged across the exploratory and the blind data sets?

Figure 1:

1). D-f: Are the PSTHs average of multiple repeated trials – if so how many? The range of firing rates expressed throughout the 3 example units is quite large – is f potentially a multi-unit? How were units isolated – did you verify with auto-correlograms? Is the PSTH offset to sound onset – that is, is 0 the time at which the sound was started? With the 5 ms ramp of the white-noise burst that means these neurons are responding almost instantly to the sound – is there latency really only 5 ms? Is there a way to increase the size of these figures – it is quite challenging to look at the PSTHs even zooming in all the way on the pdf. Although we get a feel for variance across conditions it's unclear how reliable these neurons are to the same condition. This goes for all the plots like this (fig 2 and 3)

2). G: Heat maps like this are a little deceptive – the authors sorted by azimuth and see an effect of azimuth – which is not ideal. Instead it might be more effective if the authors sorted based on A-P position, which is more in line with what question they are asking.

3). H: I'm not sure what is being plotted here – how is there error in the RF azimuth? Were there multiple distributions fit to each cell to get an error, and if so how many? Is this the location in the azimuth with the highest FR or the middle of the RF (this should be reported in the figure legend and likely the main text). Also – why report the slope of the regression in mm instead of micrometers since the axis is in micrometers? But in general this is a great figure that really shows the topographic organization of azimuth in the SC – very nice!

4). I: How symmetric are the RFs – is radius a good measure? Is there skew in the nasal RFs? Are temporal and nasal RFs different? Recent work in monkey SC has shown that visual RFs near 0 degrees in visual space are skewed while those more eccentric are less skewed (Hafed et al. Current biology 2019). Also like H – where is the error coming from – multiple fits of the k parameter (this goes for all the plots like this)? If the RFs are skewed?

Supplemental Figure 1:

1). B: Please specify in the figure legend the black dashed line is the average over 3 heads (or maybe animals is a more palatable word).

Supplemental Figure 2:

1). C: The blue bold line does not look more triphasic nor narrow than any of the other average waveform clusters to me (e.g. purple seems more triphasic and orange is just as narrow). Is this a typo? I'm not sure based on this figure that those neurons should be excluded.

Supplemental Figure 4:

1). It should be reported in the main text that the blind dataset did not reach significance (at least as a parenthetical).

Supplemental Figure 5:

1). E-f: It seems from these figures that a majority of your cells are firing less than 10 Hz in response to the stimulus – is that correct. The axis make it a little hard to judge and maybe could be clarified.

Freezing spectral cues results in the largest change to the RFs of the SC neurons

Text:

1). 53% had reproducible RFs: Why are RFS so variable?! Are these cells preferentially affected by

locomotion? Was there extra pinnae movements on some trials vs. others. This just seems very low.
2) (by simple division) – I think something like SIF / SIC would be more clear than saying by simple division.

Figure 2:

- 1). F-g: This unit's RF is hard for me to really see – is there a correlation between SI and A-P position? That is maybe cells relying on ILD are more variable than spectral relying cells.
- 2). K-o: Although I was saying Fig. 1 G should be replotted in A-P coordinates these could also use it but less important here since the question is more along the lines of how the frozen cue affect the population RF structure.
- 3). P & R: Boxplots or violin plots are better than bar plots.
- 4). Q: This is a very nice figure.
- 5). O: I can't see the red line at all – even zooming on the PDF it's very difficult to see – please thicken it

Response of SC neuron to monaural or extended ILD stimuli confirm the importance of spectral cues for encoding the azimuthal topographic map

Figure 3:

- 1). K-m: It's unclear which neural recordings this population is coming from. The heat maps have about 32 neurons but it's not in the figure legend or text where this group is coming from.
- 2). O: This heatmap has a different scale bar than the rest and should be noted with a color bar on the plot since all the other population ones range from 0 – 0.33

The nasal vs. temporal RF model:

Figure 4:

- 1). B: I found this figure to be difficult – I'm not sure why p value is on the x axis while the threshold for being a good fit (the black line) is on extended from the y axis which is neuron count?
- 2). C: This finding seems very arbitrary. The population of cells show that ILDs are important for temporal space while spectral cues are more important for nasal space. The model (which is fit to the cells) then shows that nasal space uses spectral dependent RFs while temporal space uses ILD dependent RFs. Since the model is fit to the cells – it should by definition show structure that is intrinsic to those cells.
- 3). D-g: 1). I'm not sure how you set the ITD to zero since for the ILD and Spectral cues you set the corresponding weight to 0 – so how do you freeze ITD in the model? 2). Much like C – if you have cells that have mostly spectral weights and you set that weight to 0 its arbitrary that the activity goes away and the same goes for ILD cells.
- 4). Overall – I'm entirely unsure what this model adds – it just seems to recapitulate structure from the data it was modeled on. But maybe I'm missing something about how this model is working?

Discussion:

The mouse has a topographic map of azimuthal auditory space in the SC:

- 1). "with the slope of the azimuth approaching that of the visual map in the SC" – I think it's a very interesting point that it doesn't actually map completely and this might be worth fleshing out completely. Though recent work shows that some minor differences may not be super important for decoding out position (e.g. Lee and Groh 2014).
- 2). "What is the nature of the remaining 77% of auditory responsive neurons?": I find this to be particularly troubling – what are the RFs of those 77% of neurons – just flat across the entire azimuth? Are they monotonic? If the 23% of neurons that are important for the topographic map are

embedded in a large population of auditory responsive neurons does that mean there is really a map at all? Is this just a selection bias? Are the other 77% clustered somewhere anatomically? It is certainly worth looking at this population in more detail.

Roles of ITDs, ILDs, and spectral cues in making a map of auditory space in the SC

- 1). "This property is also found in a majority of" – This is also found in the primate SC and primate IC (Lee and Groh 2014, Groh, Kelly, Underhill 2003).
- 2). "A potentially important difference between ILDs and spectral cues is that an appropriate interpretation of spectral cues requires knowledge of the original spectrum of the sound source". This is a particularly interesting point and recent work shows that these type of schema/prior can be learned extremely rapidly (though this was human work – Woods & McDermott, 2018)
- 3). "if the spectrum of the source sound is an abnormal shape or restricted to a narrow frequency band ,spectral cues will not be able to provide accurate information of the sound source location" – This is also very interesting – especially since much work in the auditory field is done with pure tones – the narrowest frequency bands. Maybe this is why spectral cues were never found to be that involved in azimuthal localization? It would be interesting to know the statistics of spectrum shapes for natural sounds.
- 4). It might be interesting to also discuss that spectral cues maybe particularly important for mammals with small heads since ITD cues seem less important for them (though many other small animals seem to evolve other systems to deal with small heads - Mason AC, Oshinsky ML, Hoy RR. 2001).
- 5). It might also be worth discussing how this is all dealing with near-field sound since the HRTF were measured from a speaker 25 cm away and how this might extend to far field sounds. Reviewers' Comments:

Reviewer #1:

Remarks to the Author:

This paper offers an interesting and technically sophisticated experimentation that generates data from a large number of cells and from a wide range of positions in space, and with a wide variety of stimulus manipulations that should offer enormous insights into the underlying mechanisms that give rise to the topographic map selectivity.

However, what is unfortunate is that the analysis feels highly incomplete. So while there are some interesting findings, for example, the finding that the topographic SC map is confirmed, and the dependence on ITD and ILD is further clarified. However, what is lacking is a more detailed consideration of the "spectral cue" and, specifically, the role of frequency tuning and inputs in determining the RFs of the cells. The authors are well aware that frequency tuning makes a difference to the way a cell responds to the ILD,s and to the spectral cues. Yet, there is no consideration of this parameter at all. For instance, it is unclear whether nasal and temporal RFs are tuned in different ways, or mixed tuning. In fact, usually spectral cues contribute to localization through the frequency tuning of the cells, and since this is important, it is unclear how the spectral cues contribute heavily to the nasal RFs. Is it simply the lack of ILD sensitivity in the nasal RFs that uncovers them? Or is the nasal RF a sculpted spectral sensitivity that is tuned to the HRTF from central locations? Curiously, ODD sigmoidal ILD dependence, and the EVEN monaural sensitivity near the nasal locations (Figs 3m,o) give rise nicely to the total RF maps (e.g., Fig3k)!

Because of the lack of any information and analysis into the frequency selectivity of the various neuronal populations, one is left with a really simple "curve fit" of a model which is not really a model in the true sense of the word, but simply a (as is well described) a simplistic description of a gaussian RF shifting left and tight as one changes the weight of the ODD to EVEN terms. One learns nothing from such a model as to how the RF's arise in the first place. I highly recommend that the authors

consider the contribution of the spectral tuning to the RF formation in different cells, and not just lump it all as the "spectral cue". As the authors point out, the spectrum contains very large implicit ILDs that are highly specific to certain frequency channels and these cannot be simply removed by equalizing the average level (as is well explained in the control experiments).

Finally, I strongly recommend changing the terms "nasal" and "temporal" to perhaps "central" and "lateral". The word "temporal" really throws off the reader into time-domain issues which are clearly not at all what the authors want.

Reviewer #2:

Remarks to the Author:

To date, most of the studies on the neural mechanisms of sound localization have largely focused on two major features: interaural timing differences (ITDs) and interaural level differences (ILDs). Here, the authors make an important contribution to the field by systematically assessing a third, largely understudied contributor to sound localization, spectral cues.

The findings are novel and interesting, the work is experimentally well conducted and of relatively broad relevance for the neuroscience community. Overall, the work seems a good fit for the journal and we support its publication, however we do have a few minor comments and we would be grateful if the authors could address them in their response.

Head related transfer function

- It would be interesting to see a graph of the difference between the contra and ipsi HRTFs (fig S1c minus fig S1d). This would give the ILD across azimuths and frequencies.
- Defining the head related transfer function is not a minor feat, the authors have done an excellent work that will greatly benefit the community. With that regard, we wonder whether the obtained HRTF could be made more easily accessible to the community to simplify the work of other groups that might want to implement a similar virtual auditory space. I guess one way would be to publish the set of coefficients for an FIR filter that replicates the transfer function.

Use of spectral cues and ILDs across the SC for auditory RFs

- The paper states that auditory RFs are formed using ILDs and spectral cues. However, it seems that in the absence of both, some neurons still maintain an auditory RF (neurons 30 to 50 in figure 2k-o). Could the authors provide an additional graphic to figure 2k-o, showing the RFs of neurons with reproducible RF under simultaneous freezing of both spectrum and ILD (and eventually ITD)? This will constitute a negative control, and these neurons should lose completely their auditory RF. If that's not the case, it would point towards another error of the virtual auditory space stimulation, in addition to the estimation of the RF size and location, and should be commented also in the methods section ("Estimation of additional systematic errors of the RF parameters"). We do appreciate that figure 3 partially addresses the above concern by saturating the ILD to a maximum.

Modelling

- Figure 4b, it should be "fraction of neurons" instead of "# of neurons"
- It is unclear how to simulate a monaural experiment with the model. The sentence "assuming that monaural stimuli have a large ILD (i.e. $ILD-RF = 1$) at every virtual location" is unclear. Practically, what changes occur to the 4 model parameters, w_1 , w_2 , k and b to simulate monaural stimulation experiments? Are these parameters unchanged and only the function $F_{ild}(x,0)$ set to a high constant?

Discussion

- As suggested in the discussion on the ILD-dependent neurons, this property is also found in the

lateral superior olive and the central nucleus of the inferior colliculus. Could you state whether there are afferent or efferent connections known between these nuclei and the SC? Could it be possible then that collicular neurons inherit Temporal ILD RF component from these nuclei?

- I'd like to further a bit the utility of the model, "useful for improving our understanding of the system and making testable predictions". It is clear that this model help understanding the combination of the two RF components for auditory collicular neurons. However, a testable prediction resulting from this model is missing. For example, the question could be raised of the origin of these two RF components. Because these two components are orthogonal, it could be that they are computed in separate afferent brain area that converge onto these neurons. Or one component could be computed locally in the SC.
- It is also missing where and how could the spectrum RF be computed. Studying the frequency tuning curve of auditory collicular neurons could help solving these question.

Reviewer #3:

Remarks to the Author:

The authors investigate the contribution of different sound localization cues to the auditory receptive field structure of neurons in the awake, mouse superior colliculus. Their major findings are:

- 1). The authors found a topographic representation of sound location in a subset of their recorded neurons along the anterior-posterior axis of the superior colliculus.
- 2). The receptive fields to these neurons relied on spectral cues and inter-aural level differences (ILD) but not inter-aural timing differences (ITD).
- 3). Furthermore the spectral cues contributed more to receptive fields concerning the nasal aspect of auditory space while ILD cues contributed more to receptive fields concerning the temporal aspect of auditory space.

The main topic of this paper is likely to be interesting to multiple fields, especially - sound localization, multisensory integration, and the burgeoning field investigating of the role the superior colliculus in cognitive functions. It is of particular interest because the authors show a role for spectral cues in azimuthal localization where they are often ignored (most research focuses on ITD and ILD of azimuthal localization). Overall, I think the evidence presented in the figures supports their major conclusions. However, I found the presentation of the data and the details about the methods concerning, what the experimenters did to collect/analyze the data is particularly opaque. I don't think the model presented in the paper adds anything substantial. I also have some minor concerns about the interpretation of the data.

General Comments:

- 1). There should be more details about the stimulus presentation – specifically, how many times were each stimuli presented at each virtual location? Were they presented in sequence? Were the frozen experiments blocked or randomly interleaved? How much time was there between sound presentations?
- 2). It is difficult to get a feel for the variance of the RF structure across the neurons. In the population plots you can get a sense but it is not explicitly stated what the error bar means (also see below for more specific comments on those plots).
- 3). Freezing of specific cues: I think this was a clever manipulation. However, the way it's discussed in the paper makes it seem like the cue vanishes and there is no more information carried via that cue. However, isn't that not the case? For instance, if you set the timing difference to zero doesn't the ITD then signal a sound source near 0 degrees. So when one cue is frozen, the sound now carries

conflicting information about sound source. I don't think this is particularly alarming issue but might warrant some attention in the discussion. It'd be interesting if the frozen experiments were blocked though – since the SC tends to weight salient/reliable stimuli more strongly than irrelevant/unreliable stimuli. If a cue no longer provides information for many trials in a row would there be plasticity in the SC to stop responding to that cue?

Section Specific Comments:

Introduction:

- 1). It seems the way the duplex theory is presented as the end-all-be-all is somewhat strawman-y. Much research over the years has called into question its strict interpretation (e.g. ref #3) – so there have been “questions about the validity of the duplex theory of sound localization” for some time now. A slight reframing with a weaker interpretation of the duplex theory might be more accurate
- 2). “The SC is an ideal brain area to study sound localization because it contains spatially tuned auditory neurons and a topographic map of auditory space.” Although in some species there is clearly an auditory map of space in the SC, in others it's quite controversial. Since this paper is claiming to be the first report of a topographic auditory map in the SC of mouse – I'm not sure saying that the SC contains a map here is ideal. Perhaps the qualifier “in some species” should be added or some combination with the next sentence would help.
- 3). “demonstrate that the mouse is a useful model to study the mechanisms of auditory processing” I'm not sure this was 1) ever in question or 2) how this paper explicitly shows how is it a useful model. It certainly does not seem to be the most relevant model to humans if that is how the authors define useful.

Methods:

Measurement of the HRTF:

- 1). You measured from 3 decapitated heads – were both male and female heads included? Since the recordings took place in both male and female mice could there be a larger bias for a particular sex? For instance if male mouse heads were used and they were on average larger than female heads (I'm not sure if there are large differences between male mice and female mice but I know it can be the case for rats).

Animal preparation for electrophysiology:

- 1). Were ear bars used in the craniotomy surgery and if so does this cause any swelling in the ear canal/pinnae that could also add to a systematic error?
- 2). “We then lower the probe until the visual response disappear” 1. How were the visual response probed? 2. There are visual responses in the deep layers of the SC so why would the visual response disappear?
- 3). How did the authors ensure the mouse was awake? Do the mice fall asleep occasionally and if so did that change the responses of the SC?

Significance test for the auditory response:

- 1). Are the data not overdispersed when the animal is still (it mentions that a possible cause of overdispersion is locomotion)?
- 2). I think this section could use some fleshing out – what is the justification for quasi-Poisson (as opposed to negative binomial – e.g. ref #34). What explicitly was done?

Function Fit to estimate the azimuth and elevation of the RFs of the neurons:

- 1). What is the justification for using the Kent distribution as opposed to another distribution (like a 2

dimensional Gaussian)?

2). Is the Kent distribution able to capture monotonic and non-monotonic RF structures equally well? This seems important – especially for the ILD analyses in the later figures.

Estimation of additional systematic errors of the RF parameters:

1). “eye movements modulate the map of auditory space to keep the visual and auditory maps aligned” This reference is from rhesus macaques where the existence of an auditory map in the SC is still debated. Additionally, the movement of the auditory RF is half of what it should be if they were to keep the auditory and visual RFs aligned – so the eye movements adjust the RFs but not to complete alignment.

2). Is there any pinna movement associated with sound presentation? Anticipation of sound? Were the left and right pinna moved independently or were they moved the same? I’m not sure looking at 100 random frames is the way to control for sound correlated pinna movements – though it seems fine for getting a range of pinna movements.

Results:

Mouse SC neurons have spatially localized auditory RFs that are topographically organized and associated figures:

Text:

1). “The neurons had a variety of temporal response patterns, with their peak response time having a bimodal distribution” I found this sentence a bit confusing – I was unsure if the bimodal distribution was talking about each individual unit having a bimodal response or the population. Adding the word population may clarify – e.g. with the population peak response time having a bimodal population.

2). I’m not entirely sure where the error is coming from in probing things like # of responsive neurons or # of neurons with a particular response latency? E.g. 77.5 +/- 0.6% of neurons had a peak response faster than 20 ms – where is the 0.6 coming from? Is this due to something intrinsic the quasi-Poisson statistics? Was this due to the blind analysis? Is that averaged across the exploratory and the blind data sets?

Figure 1:

1). D-f: Are the PSTHs average of multiple repeated trials – if so how many? The range of firing rates expressed throughout the 3 example units is quite large – is f potentially a multi-unit? How were units isolated – did you verify with auto-correlograms? Is the PSTH offset to sound onset – that is, is 0 the time at which the sound was started? With the 5 ms ramp of the white-noise burst that means these neurons are responding almost instantly to the sound – is there latency really only 5 ms? Is there a way to increase the size of these figures – it is quite challenging to look at the PSTHs even zooming in all the way on the pdf. Although we get a feel for variance across conditions it’s unclear how reliable these neurons are to the same condition. This goes for all the plots like this (fig 2 and 3)

2). G: Heat maps like this are a little deceptive – the authors sorted by azimuth and see an effect of azimuth – which is not ideal. Instead it might be more effective if the authors sorted based on A-P position, which is more in line with what question they are asking.

3). H: I’m not sure what is being plotted here – how is there error in the RF azimuth? Were there multiple distributions fit to each cell to get an error, and if so how many? Is this the location in the azimuth with the highest FR or the middle of the RF (this should be reported in the figure legend and likely the main text). Also – why report the slope of the regression in mm instead of micrometers since the axis is in micrometers? But in general this is a great figure that really shows the topographic organization of azimuth in the SC – very nice!

4). I: How symmetric are the RFs – is radius a good measure? Is there skew in the nasal RFs? Are temporal and nasal RFs different? Recent work in monkey SC has shown that visual RFs near 0 degrees in visual space are skewed while those more eccentric are less skewed (Hafed et al. Current

biology 2019). Also like H – where is the error coming from – multiple fits of the k parameter (this goes for all the plots like this)? If the RFs are skewed?

Supplemental Figure 1:

1). B: Please specify in the figure legend the black dashed line is the average over 3 heads (or maybe animals is a more palatable word).

Supplemental Figure 2:

1). C: The blue bold line does not look more triphasic nor narrow than any of the other average waveform clusters to me (e.g. purple seems more triphasic and orange is just as narrow). Is this a typo? I'm not sure based on this figure that those neurons should be excluded.

Supplemental Figure 4:

1). It should be reported in the main text that the blind dataset did not reach significance (at least as a parenthetical).

Supplemental Figure 5:

1). E-f: It seems from these figures that a majority of your cells are firing less than 10 Hz in response to the stimulus – is that correct. The axis make it a little hard to judge and maybe could be clarified.

Freezing spectral cues results in the largest change to the RFs of the SC neurons

Text:

- 1). 53% had reproducible RFs: Why are RFS so variable?! Are these cells preferentially affected by locomotion? Was there extra pinnae movements on some trials vs. others. This just seems very low.
- 2) (by simple division) – I think something like SIF / SIC would be more clear than saying by simple division.

Figure 2:

- 1). F-g: This unit's RF is hard for me to really see – is there a correlation between SI and A-P position? That is maybe cells relying on ILD are more variable than spectral relying cells.
- 2). K-o: Although I was saying Fig. 1 G should be replotted in A-P coordinates these could also use it but less important here since the question is more along the lines of how the frozen cue affect the population RF structure.
- 3). P & R: Boxplots or violin plots are better than bar plots.
- 4). Q: This is a very nice figure.
- 5). O: I can't see the red line at all – even zooming on the PDF it's very difficult to see – please thicken it

Response of SC neuron to monaural or extended ILD stimuli confirm the importance of spectral cues for encoding the azimuthal topographic map

Figure 3:

- 1). K-m: It's unclear which neural recordings this population is coming from. The heat maps have about 32 neurons but it's not in the figure legend or text where this group is coming from.
- 2). O: This heatmap has a different scale bar than the rest and should be noted with a color bar on the plot since all the other population ones range from 0 – 0.33

The nasal vs. temporal RF model:

Figure 4:

- 1). B: I found this figure to be difficult – I'm not sure why p value is on the x axis while the threshold

for being a good fit (the black line) is on extended from the y axis which is neuron count?

2). C: This finding seems very arbitrary. The population of cells show that ILDs are important for temporal space while spectral cues are more important for nasal space. The model (which is fit to the cells) then shows that nasal space uses spectral dependent RFs while temporal space uses ILD dependent RFs. Since the model is fit to the cells – it should by definition show structure that is intrinsic to those cells.

3). D-g: 1). I'm not sure how you set the ITD to zero since for the ILD and Spectral cues you set the corresponding weight to 0 – so how do you freeze ITD in the model? 2). Much like C – if you have cells that have mostly spectral weights and you set that weight to 0 its arbitrary that the activity goes away and the same goes for ILD cells.

4). Overall – I'm entirely unsure what this model adds – it just seems to recapitulate structure from the data it was modeled on. But maybe I'm missing something about how this model is working?

Discussion:

The mouse has a topographic map of azimuthal auditory space in the SC:

1). "with the slope of the azimuth approaching that of the visual map in the SC"- I think it's a very interesting point that it doesn't actually map completely and this might be worth fleshing out completely. Though recent work shows that some minor differences may not be super important for decoding out position (e.g. Lee and Groh 2014).

2). "What is the nature of the remaining 77% of auditory responsive neurons?": I find this to be particularly troubling – what are the RFs of those 77% of neurons – just flat across the entire azimuth? Are they monotonic? If the 23% of neurons that are important for the topographic map are embedded in a large population of auditory responsive neurons does that mean there is really a map at all? Is this just a selection bias? Are the other 77% clustered somewhere anatomically? It is certainly worth looking at this population in more detail.

Roles of ITDs, ILDs, and spectral cues in making a map of auditory space in the SC

1). "This property is also found in a majority of" – This is also found in the primate SC and primate IC (Lee and Groh 2014, Groh, Kelly, Underhill 2003).

2). "A potentially important difference between ILDs and spectral cues is that an appropriate interpretation of spectral cues requires knowledge of the original spectrum of the sound source". This is a particularly interesting point and recent work shows that these type of schema/prior can be learned extremely rapidly (though this was human work – Woods & McDermott, 2018)

3). "if the spectrum of the source sound is an abnormal shape or restricted to a narrow frequency band ,spectral cues will not be able to provide accurate information of the sound source location" – This is also very interesting – especially since much work in the auditory field is done with pure tones – the narrowest frequency bands. Maybe this is why spectral cues were never found to be that involved in azimuthal localization? It would be interesting to know the statistics of spectrum shapes for natural sounds.

4). It might be interesting to also discuss that spectral cues maybe particularly important for mammals with small heads since ITD cues seem less important for them (though many other small animals seem to evolve other systems to deal with small heads - Mason AC, Oshinsky ML, Hoy RR. 2001).

5). It might also be worth discussing how this is all dealing with near-field sound since the HRTF were measured from a speaker 25 cm away and how this might extend to far field sounds.

Rebuttal letter for “Spectral cues are necessary to encode azimuthal auditory space in the mouse superior colliculus” under revision at Nature Communications.

Dear Reviewers,

Please find the resubmission of our manuscript entitled “Spectral cues are necessary to encode azimuthal auditory space in the mouse superior colliculus”. We appreciate all the reviewers for their thoughtful reviews, and support on the significance of our results. As you will see, their suggestions have greatly improved the quality of the manuscript. The most significant change in this revision is the analysis of the spectrotemporal receptive fields (STRFs) of the SC auditory neurons which identified a good correspondence between the frequency tuning properties of the neurons and the head-related transfer functions (HRTFs) of the direction that the neuron is tuned to. These results are summarized in Fig. 5 and the corresponding section of the revised manuscript. We also removed the section on modeling as suggested by the reviewers.

Below are our point by point answers to the comments of each reviewer. The comments from the reviewers are written in *blue italic font* and our responses are written in black regular font. Changes made in the text are written in *red regular font*.

Reviewer #1 (Remarks to the Author):

This paper offers an interesting and technically sophisticated experimentation that generates data from a large number of cells and from a wide range of positions in space, and with a wide variety of stimulus manipulations that should offer enormous insights into the underlying mechanisms that give rise to the topographic map selectivity.

However, what is unfortunate is that the analysis feels highly incomplete. So while there are some interesting findings, for example, the finding that the topographic SC map is confirmed, and the dependence on ITD and ILD is further clarified. However, what is lacking is a more detailed consideration of the “spectral cue” and, specifically, the role of frequency tuning and inputs in determining the RFs of the cells. The authors are well aware that frequency tuning makes a difference to the way a cell responds to the ILD,s and to the spectral cues. Yet, there is no consideration of this parameter at all. For instance, it is unclear whether nasal and temporal RFs are tuned in different ways, or mixed tuning. In fact, usually spectral cues contribute to localization through the frequency tuning of the cells, and since this is important, it is unclear how the spectral cues contribute heavily to the nasal RFs. Is it simply the lack of ILD sensitivity in the nasal RFs that uncovers them? Or is the nasal RF a sculpted spectral sensitivity that is tuned to the HRTF from central locations? Curiously, ODD sigmoidal ILD dependence, and the

EVEN monaural sensitivity near the nasal locations (Figs 3m,o) give rise nicely to the total RF maps (e.g., Fig3k)!

To address the reviewer's concern, we conducted additional experiments that characterize the frequency tuning of the neurons with localized receptive fields. The results of these new experiments are summarized in the new section, "Determining the spectral pattern associated with spatial RFs" and the corresponding figure (Fig. 5) of the new manuscript.

Because of the lack of any information and analysis into the frequency selectivity of the various neuronal populations, one is left with a really simple "curve fit" of a model which is not really a model in the true sense of the word, but simply a (as is well described) a simplistic description of a gaussian RF shifting left and right as one changes the weight of the ODD to EVEN terms. One learns nothing from such a model as to how the RF's arise in the first place. I highly recommend that the authors consider the contribution of the spectral tuning to the RF formation in different cells, and not just lump it all as the "spectral cue". As the authors point out, the spectrum contains very large implicit ILD's that are highly specific to certain frequency channels and these cannot be simply removed by equalizing the average level (as is well explained in the control experiments).

We understand the reviewer's point that the proposed model was too simple to add new insights into how RFs arise in the first place. We decided to remove this section from the manuscript.

Finally, I strongly recommend changing the terms "nasal" and "temporal" to perhaps "central" and "lateral". The word "temporal" really throws off the reader into time-domain issues which are clearly not at all what the authors want.

We replaced the term "nasal" to "frontal" and "temporal" with "lateral" in the new manuscript.

Reviewer #2 (Remarks to the Author):

To date, most of the studies on the neural mechanisms of sound localization have largely focused on two major features: interaural timing differences (ITDs) and interaural level differences (LTDs). Here, the authors make an important contribution to the field by systematically assessing a third, largely understudied contributor to sound localization, spectral cues. The findings are novel and interesting, the work is experimentally well conducted and of relatively broad relevance for the neuroscience community. Overall, the work seems a good fit for the journal and we support its publication, however we do have a few minor comments and we would be grateful if the authors could address them in their response.

Head related transfer function

- It would be interesting to see a graph of the difference between the contra and ipsi HRTFs (fig S1c minus fig S1d). This would give the ILD across azimuths and frequencies.*

We added the requested figure as Fig. 5k. We also changed the format of the HRTF figures (moved from old Supplementary Fig. 1b, c to new Fig. 5g, h). They are now characterized as the

gain relative to a free-field measurement (instead of having an arbitrary baseline) and use a different colormap.

- *Defining the head related transfer function is not a minor feat, the authors have done an excellent work that will greatly benefit the community. With that regard, we wonder whether the obtained HRTF could be made more easily accessible to the community to simplify the work of other groups that might want to implement a similar virtual auditory space. I guess one way would be to publish the set of coefficients for an FIR filter that replicates the transfer function.* We will post the HRTFs of three CBA/CaJ mice and the explanation of how they can be used to a freely accessible online data repository (figshare) when this manuscript is accepted.

Use of spectral cues and ILDs across the SC for auditory RFs

- *The paper states that auditory RFs are formed using ILDs and spectral cues. However, it seems that in the absence of both, some neurons still maintain an auditory RF (neurons 30 to 50 in figure 2k-o). Could the authors provide an additional graphic to figure 2k-o, showing the RFs of neurons with reproducible RF under simultaneous freezing of both spectrum and ILD (and eventually ITD)? This will constitute a negative control, and these neurons should lose completely their auditory RF. If that's not the case, it would point towards another error of the virtual auditory space stimulation, in addition to the estimation of the RF size and location, and should be commented also in the methods section ("Estimation of additional systematic errors of the RF parameters"). We do appreciate that figure 3 partially addresses the above concern by saturating the ILD to a maximum.*

In the experiment for Fig. 3 (Fig. 2 in the old manuscript), we did not freeze ILDs and spectral cues simultaneously. Fig. 3 m-o represents the RFs when only one of the cues is frozen. Currently, we do not have data in which we froze both ILDs and spectral cues, but we did use monaural stimuli (in which ILDs are always at its maximum, as noted by the reviewer) that also have its spectrum flattened in Fig 4e, j. (Fig 3e, j in the old manuscript). In the previous manuscript, we did not show the population response to this stimulus, but in the new manuscript, we plotted it in Fig. 4n, which shows no tuned neurons. Although it is not exactly ILD and spectral cue freezing, this serves the purpose of the negative control that the reviewer suggested.

Modelling

- *Figure 4b, it should be "fraction of neurons" instead of "# of neurons"*
- *It is unclear how to simulate a monaural experiment with the model. The sentence "assuming that monaural stimuli have a large ILD (i.e. ILD-RF = 1) at every virtual location" is unclear. Practically, what changes occur to the 4 model parameters, w_1 , w_2 , k and b to simulate monaural stimulation experiments? Are these parameters unchanged and only the function $F_{ild}(x,0)$ set to a high constant?*

The reviewer's interpretation of this experiment is correct. However, based on comments by the reviewers, we decided to remove this section from the manuscript as it does not add many insights.

Discussion

- *As suggested in the discussion on the ILD-dependent neurons, this property is also found in the lateral superior olive and the central nucleus of the inferior colliculus. Could you state whether there are afferent or efferent connections known between these nuclei and the SC? Could it be possible then that collicular neurons inherit Temporal ILD RF component from these nuclei?*

We added an explanation of the synaptic relations between these nuclei as follows.

“This property is also found in the majority of neurons in **the auditory nuclei that process ILDs, namely** the lateral superior olive and **its afferent**, the central nucleus of the inferior colliculus (IC)²⁶. **The SC receives indirect input from these nuclei through the external nucleus of the IC and the nucleus of the brachium of the IC²⁷.**”

- *I'd like to further a bit the utility of the model, “useful for improving our understanding of the system and making testable predictions”. It is clear that this model help understanding the combination of the two RF components for auditory collicular neurons. However, a testable prediction resulting from this model is missing. For example, the question could be raised of the origin of these two RF components. Because these two components are orthogonal, it could be that they are computed in separate afferent brain area that converge onto these neurons. Or one component could be computed locally in the SC.*

We agree that the model is useful for understanding our results but based on the comments of the reviewers we decided to remove the section of modeling from the manuscript. We do think it is possible that SC neurons receive inputs from different brainstem nuclei; we are actively pursuing this question.

- *It is also missing where and how could the spectrum RF be computed. Studying the frequency tuning curve of auditory collicular neurons could help solving these question.*

We thank the reviewer for this suggestion. We performed additional recordings using stimuli designed to determine the spectrotemporal receptive fields of neurons that exhibit a spatial RF. The frequency tunings of the frontal neurons indeed agree well with the frontal HRTFs. The new results are displayed in Fig. 5 and the corresponding section in Results.

Reviewer #3 (Remarks to the Author):

The authors investigate the contribution of different sound localization cues to the auditory receptive field structure of neurons in the awake, mouse superior colliculus. Their major findings are:

- 1). *The authors found a topographic representation of sound location in a subset of their recorded neurons along the anterior-posterior axis of the superior colliculus.*
- 2). *The receptive fields to these neurons relied on spectral cues and inter-aural level differences (ILD) but not inter-aural timing differences (ITD).*

3). Furthermore the spectral cues contributed more to receptive fields concerning the nasal aspect of auditory space while ILD cues contributed more to receptive fields concerning the temporal aspect of auditory space.

The main topic of this paper is likely to be interesting to multiple fields, especially - sound localization, multisensory integration, and the burgeoning field investigating of the role the superior colliculus in cognitive functions. It is of particular interest because the authors show a role for spectral cues in azimuthal localization where they are often ignored (most research focuses on ITD and ILD of azimuthal localization). Overall, I think the evidence presented in the figures supports their major conclusions. However, I found the presentation of the data and the details about the methods concerning, what the experimenters did to collect/analyze the data is particularly opaque. I don't think the model presented in the paper adds anything substantial. I also have some minor concerns about the interpretation of the data.

General Comments:

1). There should be more details about the stimulus presentation – specifically, how many times were each stimuli presented at each virtual location? Were they presented in sequence? Were the frozen experiments blocked or randomly interleaved? How much time was there between sound presentations?

We are sorry that we did not place this important information in the original manuscript. The following description of the stimuli was added to include the points that the reviewer addressed.

“The stimulus was presented every 2 seconds, and repeated 30 times per direction (total duration was 85 minutes).”

“Stimuli with frozen cues were randomly interleaved so that the long-term change of the recording condition does not influence the differences of the RFs. For this experiment, we repeated stimuli 20 times per direction instead of 30 times to shorten the total duration (with 5 conditions, 85 directions, 2 s intervals, the total duration was 4.7 hours).”

2). It is difficult to get a feel for the variance of the RF structure across the neurons. In the population plots you can get a sense but it is not explicitly stated what the error bar means (also see below for more specific comments on those plots).

The error bars of many of the fits are derived from the function fit procedure. (Hessian matrix of the likelihood function gives the error matrix of the parameters. The error of each parameter is calculated by taking a square root of the diagonal element of the error matrix.). We stated how the errors are derived when they appear for the first time as follows.

“The error of each parameter was estimated from the Hessian matrix of the likelihood function.”

3). Freezing of specific cues: I think this was a clever manipulation. However, the way it's discussed in the paper makes it seem like the cue vanishes and there is no more information carried via that cue. However, isn't that not the case? For instance, if you set the timing difference to zero doesn't the ITD then signal a sound source near 0 degrees. So when one cue is frozen, the sound now carries conflicting information about sound source. I don't think this is

particularly alarming issue but might warrant some attention in the discussion. It'd be interesting if the frozen experiments were blocked though – since the SC tends to weight salient/reliable stimuli more strongly than irrelevant/unreliable stimuli.

This is a good point. It is true that the frozen cue does not vanish and can give information to neurons. We added a paragraph about this point in Methods.

“Note that freezing a sound localization cue does not mean erasing the cue. The frozen cue is still present at every location and may give information that contradicts other cues. For example, zero ILDs are consistent with a sound that comes from the midline, and some neurons may respond in a non-natural manner when this is combined with spectral cues that vary in the virtual space.”

If a cue no longer provides information for many trials in a row would there be plasticity in the SC to stop responding to that cue?

We do not know the answer to this question because we tried to avoid the adaptation of neurons by interleaving the stimuli. Understanding the plasticity of the auditory map is an avenue we will pursue, but this is beyond the scope of the current study.

Section Specific Comments:

Introduction:

1). It seems the way the duplex theory is presented as the end-all-be-all is somewhat strawman-y. Much research over the years has called into question its strict interpretation (e.g. ref #3) – so there have been “questions about the validity of the duplex theory of sound localization” for some time now. A slight reframing with a weaker interpretation of the duplex theory might be more accurate

We agree with the reviewer about this point. We deleted the statement *“leading to questions about the validity of the duplex theory of sound localization”* to make the interpretation more accurate.

2). “The SC is an ideal brain area to study sound localization because it contains spatially tuned auditory neurons and a topographic map of auditory space.” Although in some species there is clearly an auditory map of space in the SC, in others it’s quite controversial. Since this paper is claiming to be the first report of a topographic auditory map in the SC of mouse – I’m not sure saying that the SC contains a map here is ideal. Perhaps the qualifier “in some species” should be added or some combination with the next sentence would help.

We changed the text to the following.

“We have chosen to study the SC because it contains spatially tuned auditory neurons and, in some species, a topographic map of auditory space has been observed.”

3). “demonstrate that the mouse is a useful model to study the mechanisms of auditory processing” I’m not sure this was 1) ever in question or 2) how this paper explicitly shows how is it a useful model. It certainly does not seem to be the most relevant model to humans if that is how the authors define useful.

We deleted that part of the text.

Methods:

Measurement of the HRTF:

1). *You measured from 3 decapitated heads – were both male and female heads included? Since the recordings took place in both male and female mice could there be a larger bias for a particular sex? For instance if male mouse heads were used and they were on average larger than female heads (I'm not sure if there are large differences between male mice and female mice but I know it can be the case for rats).*

The mice we used for HRTF measurement were all females. We have records of the ear dimension (long axis and short axis) for both sexes. Although males were significantly heavier than females, we did not observe any significant differences between the ear dimensions. We also added a figure that shows the topographic map separately for males and females (Supplementary Fig. 8) that does not show a difference between sexes. Therefore, we concluded that the difference between males and females is not a concern for this study. We added the following descriptions in Methods.

“The HRTFs were collected from female mice, but stimuli based on them were presented to both male and female mice. We did not see a difference in the topographic map parameters between male and female groups (Supplementary Fig. 8). In addition, even though their average body weights were significantly different (male: 28.2 ± 0.7 g (N = 6), female: 21.3 ± 1.5 g (N = 6), $p = 2e-3$ (using t-test)), the ear sizes were not significantly different. (The numbers represent [right ear long axis, right ear short axis, left ear long axis, left ear short axis]. male: [13.0 ± 0.3 , 7.8 ± 0.2 , 13.2 ± 0.2 , 7.5 ± 0.1]; female: [12.6 ± 0.3 , 7.9 ± 0.3 , 12.6 ± 0.3 , 7.9 ± 0.2]; $p = [0.39, 0.90, 0.12, 0.26]$)”

Animal preparation for electrophysiology:

1). *Were ear bars used in the craniotomy surgery and if so does this cause any swelling in the ear canal/pinnae that could also add to a systematic error?*

Ear bars were not used to avoid any damage to the ear. We added a description in the corresponding section in Methods.

“We did not use ear bars to attach a head plate or to perform craniotomy in order to avoid damaging the ears.”

2). *“We then lower the probe until the visual response disappear” 1. How were the visual response probed? 2. There are visual responses in the deep layers of the SC so why would the visual response disappear?*

The reviewer is correct about the visual responses that do not completely disappear in the deep layers. What we meant here was the strong multi-unit activity that is seen only in the superficial layers of the SC. We changed the text to the following.

“We then lower the probe until the **strong multi-unit** visual responses disappear.”

3). How did the authors ensure the mouse was awake? Do the mice fall asleep occasionally and if so did that change the responses of the SC?

We only had a rotary encoder to monitor the behavior of the mouse and separated into running and stationary states based on locomotion speed. We did not ensure that the mouse was awake or asleep in the stationary period. Because we see the similar topographic organization of the auditory map in both running and stationary states (Supplementary Fig. 6), it is unlikely that if the mouse was asleep during part of the recording session it changes the main conclusions of this manuscript.

However, observing the differences between awake and sleeping states is a good idea and the result could tell us the effect of alertness/attention. We intend to video the mouse during the experiment in the future.

Significance test for the auditory response:

1). Are the data not overdispersed when the animal is still (it mentions that a possible cause of overdispersion is locomotion)?

We calculated the average overdispersion parameter for all the neurons during overall, stationary and running periods. Indeed, the overdispersion parameters during both stationary states and running states were smaller than the overall overdispersion parameter. However, the values were not close to one, suggesting that running is one of the sources, but not the sole source of overdispersion. We added a description of this point in Methods as follows.

“(The average overdispersion parameters of all neurons was 5.61 ± 0.10 . These parameters during stationary and running periods were 4.99 ± 0.08 and 5.11 ± 0.10 , respectively, indicating that locomotion is one of the sources, but not a sole source of overdispersion.)”

2). I think this section could use some fleshing out – what is the justification for quasi-Poisson (as opposed to negative binomial – e.g. ref #34). What explicitly was done?

We rewrote the section as follows.

“We used quasi-Poisson statistics for significance tests of the auditory responses of individual neurons³⁴. Simple Poisson statistics was not sufficient because the post-stimulus firing rate typically had a larger trial-by-trial variance than that expected from Poisson statistics (overdispersion) due to factors such as bursting of the neural activity and/or locomotion/movement of the animal. These additional fluctuations can cause increased false positives. (The overdispersion parameter was estimated by the variance of the spike count divided by the mean, which should be 1 if the neuron is Poissonian. The average of the overdispersion parameters of all neurons was 5.61 ± 0.10 . These parameters during stationary and running periods were 4.99 ± 0.08 and 5.11 ± 0.10 , respectively, indicating that locomotion is one of the sources, but not a sole source of overdispersion.)

To determine the significance of the response, we first estimated an overdispersion parameter and considered a response to be significant if the p-value of the neuron’s spike count is below 0.001 ($p = 1 - \text{CDF}(N)$, where CDF is a cumulative distribution function of the quasi-Poisson distribution and N is the spike count of the neuron). We have chosen quasi-Poisson over a negative binomial distribution (another distribution that permits

overdispersion) because of its simplicity. We do not expect a large difference in results because of the choice between these two distributions.”

Function Fit to estimate the azimuth and elevation of the RFs of the neurons:

1). What is the justification for using the Kent distribution as opposed to another distribution (like a 2 dimensional Gaussian)?

The Kent distribution gives a Gaussian-like peak in the two-dimensional spherical coordinate system. The two-dimensional Gaussian function does not work well in the spherical coordinate system because of the warping of the variables near the pole. (Same as the issue of making a flat world map without distortion). We added the following sentence in the text.

“We have chosen the Kent distribution over the two-dimensional Gaussian distribution because we consider a large range of angles in the directional space in which the nonlinearity of the coordinate system should be taken into account.”

2). Is the Kent distribution able to capture monotonic and non-monotonic RF structures equally well? This seems important – especially for the ILD analyses in the later figures.

Because we used the two-dimensional spherical coordinate system, the firing rate that goes up with a certain azimuth has to go down at some point in order to satisfy the periodic boundary condition $f(\theta, \phi) = f(\theta, \phi + 2\pi)$. Therefore, there is no “monotonic” RF in the spherical coordinate system. If a neuron has a monotonic response to ILDs, the neuron should have a spatially restricted RF near the direction in which the ILD is maximized (see Supplementary Fig. 1c). The RFs of these neurons can be captured as well as other localized RFs by the Kent distribution.

Estimation of additional systematic errors of the RF parameters:

1). “eye movements modulate the map of auditory space to keep the visual and auditory maps aligned” This reference is from rhesus macaques where the existence of an auditory map in the SC is still debated. Additionally, the movement of the auditory RF is half of what it should be if they were to keep the auditory and visual RFs aligned – so the eye movements adjust the RFs but not to complete alignment.

We thank the reviewer for pointing out potentially misleading use of the reference. We corrected it to a more accurate sentence as follows. “In primates, eye movement influences the auditory receptive field of the SC neurons⁴⁴. However, this property has not been characterized in mice and the range of the mouse eye movement is small.”

2). Is there any pinna movement associated with sound presentation? Anticipation of sound? Were the left and right pinna moved independently or were they moved the same? I’m not sure looking at 100 random frames is the way to control for sound correlated pinna movements – though it seems fine for getting a range of pinna movements.

We observed a small movement of the ear after the stimulus presentation. We do not see a movement that seems to anticipate the sound. The ear movement could be dependent on the properties of the sound (ILD or intensity). As the reviewer pointed out, the goal of this analysis is to estimate the systematic error due to the general range of the ear positions during the experiment and associating the ear movement with the stimulus is out of the scope of this

manuscript. Because we are focusing on analyzing the spikes within 20 ms after the stimulus onset, this stimulus-induced ear movement is unlikely to change the main conclusion of this manuscript.

We are interested in the correlation between ear/eye movements and locomotion with auditory responses. We plan to implement more sophisticated tracking software that will allow us to perform a cross-correlation analysis of movements and firing to determine this.

Results:

Mouse SC neurons have spatially localized auditory RFs that are topographically organized and associated figures:

Text:

1). *“The neurons had a variety of temporal response patterns, with their peak response time having a bimodal distribution” I found this sentence a bit confusing – I was unsure if the bimodal distribution was talking about each individual unit having a bimodal response or the population. Adding the word population may clarify – e.g. with the population peak response time having a bimodal population.*

To avoid confusion, we changed it to the following.

“The neurons had a variety of temporal response patterns (Supplementary Fig. 3a–d), **and the peak response time for the population showed a bimodal distribution (Supplementary Fig. 3e).**”

2). *I’m not entirely sure where the error is coming from in probing things like # of responsive neurons or # of neurons with a particular response latency? E.g. 77.5 +/- 0.6% of neurons had a peak response faster than 20 ms – where is the 0.6 coming from? Is this due to something intrinsic the quasi-Poisson statistics? Was this due to the blind analysis? Is that averaged across the exploratory and the blind data sets?*

We added the following description in Methods as the **“Error estimation for categorical numbers”** section, and a reference to this description for the first time this type of error appears in the main text. The reported results are based on both the exploratory and blinded datasets unless noted otherwise.

“When we evaluate categorical populations such as a fraction of neurons with a fast response, we assumed a binomial distribution. If the probability that a neuron is in a category is p , its error is estimated by $\sqrt{p(1-p)/N}$, where N is the total number of neurons.”

Figure 1:

1). *D-f: Are the PSTHs average of multiple repeated trials – if so how many? The range of firing rates expressed throughout the 3 example units is quite large – is f potentially a multi-unit? How were units isolated – did you verify with auto-correlograms? Is the PSTH offset to sound onset – that is, is 0 the time at which the sound was started? With the 5 ms ramp of the white-noise burst that means these neurons are responding almost instantly to the sound – is there latency really only 5 ms? Is there a way to increase the size of these figures – it is quite challenging to look at the PSTHs even zooming in all the way on the pdf. Although we get a feel for variance*

across conditions it's unclear how reliable these neurons are to the same condition. This goes for all the plots like this (fig 2 and 3)

We apologize for not including important information. The stimuli are repeated 30 times. This number is now included in the figure legend and Methods. We also added details of our spike-sorting in Methods. We use multiple measures (refractory period violation in the auto-correlation function, isolation distance, L-ratio, cross-correlation, electrophysiological images) to make sure the detected neurons were single isolated units. For details, please refer to the articles cited in the new sub-section, "Spike-sorting" under "Data analysis" in Methods. To give a sense of how the response varies across trials, we gave a full raster plot for the three neurons that were shown in Fig. 2a-c (Fig. 1d-f in the old manuscript) in Supplementary Fig. 4.

2). G: Heat maps like this are a little deceptive – the authors sorted by azimuth and see an effect of azimuth – which is not ideal. Instead it might be more effective if the authors sorted based on A-P position, which is more in line with what question they are asking.

Fig. 2d (Fig. 1g in the old manuscript) is supposed to show the shape of the tuning curves and whether the distribution of the RF azimuths is continuous or not. We ask whether there is a topographic organization in Fig. 2e (Fig. 1h in the old manuscript). We added more explanation on this point as follows.

"We found that the SC neurons have **bell-shaped tuning curves with** a continuous distribution of preferred azimuths in the horizontal plane (Fig. 2d), with the RF azimuth linearly related to the neurons' A–P location (Fig. 2e)."

3). H: I'm not sure what is being plotted here – how is there error in the RF azimuth? Were there multiple distributions fit to each cell to get an error, and if so how many? Is this the location in the azimuth with the highest FR or the middle of the RF (this should be reported in the figure legend and likely the main text). Also – why report the slope of the regression in mm instead of micrometers since the axis is in micrometers? But in general this is a great figure that really shows the topographic organization of azimuth in the SC – very nice!

We added the following explanations in the figure legend.

"Each blue dot represents the auditory RF azimuth (**center of the Kent distribution**) and the A–P position of an individual neuron (the error bars represent the statistical errors **derived from the Kent distribution fits**; **they do not** include the systematic errors discussed in Methods);"

Also, we added this sentence to the corresponding Method section.

"The error of each parameter was estimated from the Hessian matrix of the likelihood function."

We corrected the plot so that now it uses mm consistently instead of μm .

4). I: How symmetric are the RFs – is radius a good measure? Is there skew in the nasal RFs? Are temporal and nasal RFs different? Recent work in monkey SC has shown that visual RFs near 0 degrees in visual space are skewed while those more eccentric are less skewed (Hafed et al. Current biology 2019). Also like H – where is the error coming from – multiple fits of the k parameter (this goes for all the plots like this)? If the RFs are skewed?

The skewness of the RFs is characterized by the β parameter in the Kent distribution fit. We indeed observed a significant negative slope between the A-P position and β (-0.09 ± 0.01

mm⁻¹). However, we did not include this result in the revised manuscript because this value was small and only remotely related to the major findings of this article.

Again, the error of the parameters was the result of the function fit as described above.

Supplemental Figure 1:

1). B: Please specify in the figure legend the black dashed line is the average over 3 heads (or maybe animals is a more palatable word).

It represents the average difference between pairs of HRTFs. We added the following text in the caption.

“The black dashed line indicates the RMS difference of the HRTFs between different animals (3 pairs from 3 animals).”

Supplemental Figure 2:

1). C: The blue bold line does not look more triphasic nor narrow than any of the other average waveform clusters to me (e.g. purple seems more triphasic and orange is just as narrow). Is this a typo? I'm not sure based on this figure that those neurons should be excluded.

We agree with the reviewer that the conclusion of this figure might have been a little difficult to draw solely based on this figure. We added explanations of how we decided that the blue cluster is axonal signals, showing an example neuron that exhibits both of the cell-body signal and the axonal signal with time delay. The revised explanation is written in the legend of Supplementary Fig. 2b-d (pasted below).

“b: A 3 dimensional scatter plot of the waveforms in the space of the 3 most important principal components. The color of the dots indicates the result of the clustering analysis based on a mixture of Gaussians model. **Note that the purple cluster is larger and sparser than the others (the determinant of the Gaussian covariance matrix (i.e. volume) of the four clusters were 7.6×10^{-5} , 7.1×10^{-5} , 1.7×10^{-6} , and 1.3×10^{-2} , for the yellow, red, blue, and purple clusters, respectively, showing that the purple cluster is more than two orders of magnitude larger than the second-largest cluster). It indicates that the purple cluster is fitting to outliers. These outliers contain mostly biphasic spikes with opposite polarity (a peak followed by a trough) that are detected at the dendrites near the soma. Therefore we did not exclude this cluster from the analysis.**

c: Same as (a) with each waveform colored with the corresponding cluster color in (b). The bold solid lines indicate the average waveform of each cluster. Of these 4 **traces**, the blue color indicates the axonal signal, based on the **shape of the waveform that is consistent with an axonal signal (see (d))**. These neurons were excluded from the analysis.

d: **Example waveforms of the somatic signals and axonal signals detected from one neuron. The circles on the left panel indicate the amplitudes of the spike of this neuron detected by electrodes at the corresponding locations. The red circle indicates the electrode where this neuron was identified in spike-sorting. The average voltage traces of this neuron at two electrodes were compared on the right panels. The spike at Electrode 1 appears 0.1 ms earlier than that at Electrode 2, indicating that the spike on Electrode 2 is an axonal signal of the neuron whose cell body is located near Electrode 1. With such observations, we concluded that**

a spike with a right shoulder (yellow and orange traces in (c)) is a cell body signal, and a narrow and triphasic signal with a weaker right shoulder (blue traces in (c)) is an axonal signal.”

Supplemental Figure 4:

1). *It should be reported in the main text that the blind dataset did not reach significance (at least as a parenthetical).*

We added the following text to the reference to Supplementary Fig. 5.

“(the slope was not significant in the blinded dataset. See Supplementary Fig. 5)”

Supplemental Figure 5:

1). *E-f: It seems from these figures that a majority of your cells are firing less than 10 Hz in response to the stimulus – is that correct. The axis make it a little hard to judge and maybe could be clarified.*

We put grid lines to make the figure more visible. It is correct that the majority of the neurons are less than 10 Hz. These are all the neurons that have a significant response to auditory stimuli, but are not limited to those that have a localized RF. Besides, the numbers should not be directly compared to what’s indicated in Fig. 2a-c (old Fig. 1d-f) because Fig. 2a-c uses 5-ms bins while these Supplementary Fig. e-g uses the entire duration of the corresponding periods (1 s, 15 ms, and 180 ms, respectively).

Freezing spectral cues results in the largest change to the RFs of the SC neurons

Text:

1). *53% had reproducible RFs: Why are RFS so variable?! Are these cells preferentially affected by locomotion? Was there extra pinnae movements on some trials vs. others. This just seems very low.*

The reason is that many of the neurons do not have very reliable responses, and many of the detected localized receptive fields were near the detection threshold. In particular, because we repeated the stimulus 20 times instead of 30 in order to keep a reasonable stimulus duration, more neurons were near the detection threshold in this experiment than that for Fig. 2 (Fig. 1 in old manuscript).

2) *(by simple division) – I think something like Sf / Sfc would be more clear than saying by simple division.*

We changed the text as follows.

“The SI value of the control dataset was used to normalize the SI values of the frozen cue results (normalized SI: $NSI = SI / SI_{control}$).”

Figure 2:

1). *F-g: This unit’s RF is hard for me to really see – is there a correlation between SI and A-P position? That is maybe cells relying on ILD are more variable than spectral relying cells.*

Although there were many good examples of cells that use both ILDs and spectral cues with different weights (as indicated in Fig. 3q), we did not find a very good example of pure

ILD-dependent neurons. We did not find a correlation between SI_{control} and A-P position ($r = 0.095$, $p = 0.5$).

2). K-o: *Although I was saying Fig. 1 G should be replotted in A-P coordinates these could also use it but less important here since the question is more along the lines of how the frozen cue affect the population RF structure.*

As we noted above, the purpose of this figure is to show a continuous distribution of the RFs. We kept these figures as they are.

3). P & R: *Boxplots or violin plots are better than bar plots.*

We changed them to violin plots.

4). Q: *This is a very nice figure.*

5). O: *I can't see the red line at all – even zooming on the PDF it's very difficult to see – please thicken it*

We thickened the line.

Response of SC neuron to monaural or extended ILD stimuli confirm the importance of spectral cues for encoding the azimuthal topographic map

Figure 3:

1). K-m: *It's unclear which neural recordings this population is coming from. The heat maps have about 32 neurons but it's not in the figure legend or text where this group is coming from.*

We are sorry for not including these details. We added the following sentence in the main text.

“In this experiment, we recorded from 869 neurons from 5 mice, and found that $59.5 \pm 1.6\%$ ($n = 554$) had a significant auditory response, $12.6 \pm 1.4\%$ ($n = 70$) of these had a localized RF, and $45 \pm 6\%$ ($n = 32$) of these had a reproducible RF.”

2). O: *This heatmap has a different scale bar than the rest and should be noted with a color bar on the plot since all the other population ones range from 0 – 0.33*

We placed a color bar with a range for this figure (new Fig. 4p).

The nasal vs. temporal RF model:

Figure 4:

1). B: *I found this figure to be difficult – I'm not sure why p value is on the x axis while the threshold for being a good fit (the black line) is on extended from the y axis which is neuron count?*

2). C: *This finding seems very arbitrary. The population of cells show that ILDs are important for temporal space while spectral cues are more important for nasal space. The model (which is fit to the cells) then shows that nasal space uses spectral dependent RFs while temporal space uses ILD dependent RFs. Since the model is fit to the cells – it should by definition show structure that is intrinsic to those cells.*

3). D-g: 1). I'm not sure how you set the ITD to zero since for the ILD and Spectral cues you set the corresponding weight to 0 – so how do you freeze ITD in the model? 2). Much like C – if you have cells that have mostly spectral weights and you set that weight to 0 its arbitrary that the activity goes away and the same goes for ILD cells.

4). Overall – I'm entirely unsure what this model adds – it just seems to recapitulate structure from the data it was modeled on. But maybe I'm missing something about how this model is working?

We agree with the reviewer that this model does not add much to what's already found in previous sections. Instead, we added a figure that simply recapitulates the findings (Fig. 6).

Discussion:

The mouse has a topographic map of azimuthal auditory space in the SC:

1). *“with the slope of the azimuth approaching that of the visual map in the SC”- I think it's a very interesting point that it doesn't actually map completely and this might be worth fleshing out completely. Though recent work shows that some minor differences may not be super important for decoding out position (e.g. Lee and Groh 2014).*

We added the following text in Discussion.

“By recording from SC neurons in awake-behaving mice in response to VAS stimulation, we found that the mouse SC contains auditory neurons that form an azimuthal topographic map of sound, with the slope of azimuth approaching that of the visual map in the SC, **albeit with a small but significant difference between the slopes of the visual and auditory maps. We do not yet know whether this difference is functionally important, but as suggested in a primate study²⁰, it is possible that downstream processing can compensate for this small difference between these two maps.**”

2). *“What is the nature of the remaining 77% of auditory responsive neurons?”: I find this to be particularly troubling – what are the RFs of those 77% of neurons – just flat across the entire azimuth? Are they monotonic? If the 23% of neurons that are important for the topographic map are embedded in a large population of auditory responsive neurons does that mean there is really a map at all? Is this just a selection bias? Are the other 77% clustered somewhere anatomically? It is certainly worth looking at this population in more detail.*

The remaining neurons had a smaller BIC with a flat distribution than the Kent distribution. This indicates that the gain in the likelihood values was not sufficient to justify the additional parameters of the Kent distribution. Our interpretation of this is that the distribution is flat or simply noisy so that a function fit is not justified. Again, monotonic RFs do not exist if you consider a large enough range of the azimuth, so our Kent distribution will fit the firing rate distribution of the neurons with a large RF near the side of the animal.

One thing to note is sensitivity. The significance of the presence of response uses all 85 spatial grids altogether, while the fit considers them as individual points. The statistical power for detecting the signal is much higher for a simple significance test. So even if the neuron is considered to have a significant response, each position may not have a sufficient number of spikes to be considered as a localized RF.

However, there are also a number of cells that clearly respond to sound but do not have a spatially structured RF. It is certainly interesting to look into these neurons, but such an investigation is outside the scope of the current manuscript. We may follow up on these aspects in a separate study.

Roles of ITDs, ILDs, and spectral cues in making a map of auditory space in the SC

1). *“This property is also found in a majority of” – This is also found in the primate SC and primate IC (Lee and Groh 2014, Groh, Kelly, Underhill 2003).*

We added references to these articles.

2). *“A potentially important difference between ILDs and spectral cues is that an appropriate interpretation of spectral cues requires knowledge of the original spectrum of the sound source”. This is a particularly interesting point and recent work shows that these type of schema/prior can be learned extremely rapidly (though this was human work – Woods & McDermott, 2018)*

It is indeed interesting to test whether the spectrum of a sound source can be learned. We added a following sentence in the discussion.

“Although humans can learn the abstract structure of sound rapidly through experience³², whether such information can facilitate sound localization is not known.”

3). *“if the spectrum of the source sound is an abnormal shape or restricted to a narrow frequency band ,spectral cues will not be able to provide accurate information of the sound source location” – This is also very interesting – especially since much work in the auditory field is done with pure tones – the narrowest frequency bands. Maybe this is why spectral cues were never found to be that involved in azimuthal localization? It would be interesting to know the statistics of spectrum shapes for natural sounds.*

Indeed, early works on sound localization that used pure tones may have hindered the importance of spectral cues. However, some of the studies that used monaural stimuli and examined the horizontal plane suggested the potential importance of spectral cues in horizontal sound localization (Refs. 5 and 6).

Although it is out of the scope of the present study, we are interested in investigating the statistical properties of the natural sound.

4). *It might be interesting to also discuss that spectral cues maybe particularly important for mammals with small heads since ITD cues seem less important for them (though many other small animals seem to evolve other systems to deal with small heads - Mason AC, Oshinsky ML, Hoy RR. 2001).*

We added the following text in Discussion.

“There is a possibility that spectral cues are particularly more important for mice because they cannot utilize ITDs. Therefore, to examine the extension of our findings across species, the role of spectral cues in species that utilize ITDs for horizontal sound localization will need to be studied.”

5). *It might also be worth discussing how this is all dealing with near-field sound since the HRTF were measured from a speaker 25 cm away and how this might extend to far field sounds.*

We added the following text in Discussion

“It will also be interesting to test sound from different distances, a parameter that we did not examine in this study because higher frequency sound is preferentially dissipated by traveling a long distance in the air.”

Reviewers' Comments:

Reviewer #1:

Remarks to the Author:

I am satisfied with the revisions and responses offered by the authors and have no more reservations regarding the manuscript publication.

Reviewer #2:

Remarks to the Author:

I am satisfied with the answers given by the authors. I congratulate them on an excellent work and I recommend its publication.

Reviewer #3:

Remarks to the Author:

Comments to authors:

The authors adequately address the reviews. The point by point responses seem quite reasonable. A key change in the manuscript is the addition of figure 5 which probes the spectral sensitivity of the auditory responsive neurons in the superior colliculus. The authors report that neurons that mostly rely on spectral cues or ILDs for their spatial RFs are differentially sensitive to different frequency ranges and this aligns well with the HRTF. In addition to strengthening the auditory map finding, the mapping of STRFs in the mouse superior colliculus adds to the novelty of the manuscript since very little work is done on auditory responses in the superior colliculus let alone their frequency selectivity. Overall, the approach to probing the STRFs is well done and strengthens the manuscript major findings and novelty.

I only have a few, minor comments on the revised manuscript:

Supplementary figures:

- 1). Figure 2d: Typo – “yellow and orange trances” I believe traces is meant here.
- 2). Figure 6i: I missed this in my first comments – is the slope of the topographic auditory map significantly different than the slope of the visual map in the running condition? The regression line seems to be a very good fit for the red data as well in this figure.
- 3). Figure 8: What are the N for each group (male and female).

Main Text/Point-to-point Response:

- 1). Lines 296-301 “This property is also found..... 79 sensitive ILD neurons”. The argument of Refs 20 and 28 seems to be that most of the neurons are monotonic (better fit by sigmoid functions) in the frontal hemifield of space. Though the citation in this discussion uses them for evidence of non-monotonicity.

Point-by-point responses to reviews for “Spectral cues are necessary to encode azimuthal auditory space in the mouse superior colliculus” under revision at Nature Communications.

Below are our point by point answers to the comments of each reviewer. The comments from the reviewers are written in *blue italic font* and our responses are written in black regular font. Changes made in the text are written in **red regular font**.

Reviewer #1 (Remarks to the Author):

I am satisfied with the revisions and responses offered by the authors and have no more reservations regarding the manuscript publication.

Reviewer #2 (Remarks to the Author):

I am satisfied with the answers given by the authors. I congratulate them on an excellent work and I recommend its publication.

Reviewer #3 (Remarks to the Author):

Comments to authors:

The authors adequately address the reviews. The point by point responses seem quite reasonable.

A key change in the manuscript is the addition of figure 5 which probes the spectral sensitivity of the auditory responsive neurons in the superior colliculus. The authors report that neurons that mostly rely on spectral cues or ILDs for their spatial RFs are differentially sensitive to different frequency ranges and this aligns well with the HRTF. In addition to strengthening the auditory map finding, the mapping of STRFs in the mouse superior colliculus adds to the novelty of the manuscript since very little work is done on auditory responses in the superior colliculus let alone their frequency selectivity. Overall, the approach to probing the STRFs is well done and strengthens the manuscript major findings and novelty.

I only have a few, minor comments on the revised manuscript:

Supplementary figures:

1). Figure 2d: Typo – “yellow and orange trances” I believe traces is meant here.

Thank you for finding our mistake. We corrected the word in the revised manuscript.

2). *Figure 6i: I missed this in my first comments – is the slope of the topographic auditory map significantly different than the slope of the visual map in the running condition? The regression line seems to be a very good fit for the red data as well in this figure.*

The reviewer is correct. As also indicated in the main text, the overall slope of the auditory map is ~20% smaller than that of the visual map (line 99 of the merged manuscript in Rev. 1). We added the significance of the difference of the slope in the main text to clarify the observed difference between the visual and auditory maps. Although the regression line seems to fit well, the measured slope by the fit shows a significant difference.

3). *Figure 8: What are the N for each group (male and female).*

We used 6 males and 14 females in this analysis. We added these numbers in the figure legend.

Main Text/Point-to-point Response:

1). *Lines 296-301 “This property is also found..... 79 sensitive ILD neurons”. The argument of Refs 20 and 28 seems to be that most of the neurons are monotonic (better fit by sigmoid functions) in the frontal hemifield of space. Though the citation in this discussion uses them for evidence of non-monotonicity.*

To avoid potential misleading, we changed the text as follows.

Few non-monotonic neurons were found in the IC of bats²⁶ and primates²⁸ and the SC of cats²⁹ and primates²⁰; **consistent with these observations**, we found only one non-monotonic neuron out of the 79 ILD sensitive neurons.